# Stratigraphic architecture of the Belly River Group (Campanian, Cretaceous) in the plains of southern Alberta: Revisions and updates to an existing model and implications for correlating dinosaur-rich strata

**David A. Eberth** *

Royal Tyrrell Museum of Palaeontology, Drumheller, Alberta, Canada

* deberth@cciwireless.ca

## Abstract

The Upper Cretaceous (Campanian) Belly River Group (BRG) of southern Alberta has a complex internal stratigraphic architecture derived from differential geometries of its component formations that resulted from regionalized tectonic influences and shifting source areas. A full understanding of BRG architecture has been compromised heretofore by a limited understanding of subsurface data in southwestern- and southeastern-most Alberta. In this study outcrop exposures throughout southern Alberta are tied to reference well logs and subsurface cross-sections allowing a more precise understanding of BRG architecture and how it relates to well-known vertebrate fossil producing areas. Modifications to an existing stratigraphic model of the BRG show that the Oldman and the Dinosaur Park formations have reciprocal north-to-south wedge-shaped geometries and a diachronous contact that become prominently expressed south of Twp 12. The updated model also demonstrates that the Oldman Formation thickens stratigraphically up-section to the south, and that the Foremost-Oldman contact is, essentially, a datum across much of southern Alberta. Identification of the Oldman Formation in the subsurface remains based on its relatively high gamma-ray response in mudstone successions, but it is also recognized that many of its sandstones exhibit relatively low gamma-ray responses like those in underlying and overlying formations. Nomenclature and subdivisions of the Oldman Formation are revised to accommodate this updated understanding, and modifications are also made to the definition of the Judith River-Belly River discontinuity, a newly recognized surface that marks the onset of accommodation and eustatic rise in sea-level in the northern Western Interior Basin at ~76.3 Ma.

**Data Availability Statement:** All relevant data are within the paper and its Supporting Information files.

**Funding:** The author received no specific funding for this work.

**Competing interests:** The authors have declared that no competing interests exist.

## Introduction

The complex internal stratigraphic architecture of the Belly River Group (BRG) clastic wedge in the plains of southern Alberta derives from differential geometries of its component formations across its regional expression [1–9]. Until now, aspects of this complex architecture have been documented within limited stratigraphic intervals and geographic areas, creating a piecemeal understanding of the wedge, and leaving a "big picture" understanding of the wedge and its formations largely unrealized. This is unfortunate because the BRG is a significant source of Campanian-age oil, gas, and coal resources as well as dinosaur and other vertebrate fossils in the Western Interior Basin (WIB) of North America [6, 9–12]. Accordingly, accurate litho- and chronostratigraphic correlation within the BRG at numerous widely separated localities across the southern Alberta plains and beyond Alberta depends on a clear understanding of how the group's formational geometries relate to one another as well as to the units that bound the wedge (Pakowki/Lea Park and Bearpaw formations). Furthermore, advances continue to be made in our understanding of the litho- and chronostratigraphy of adjacent correlative units in Saskatchewan and Montana [e.g., 9, 13, 14] making it increasingly important that the BRG be understood in terms of both its parts and the whole to facilitate accurate correlation and meaningful interpretations of its depositional history and preservation within the northern WIB.

Reconstruction of Belly River Group architecture across southern Alberta is now possible due to a combination of location-specific studies (outcrop and subsurface) that have been conducted during the past 80 years, and the relatively recent availability of a rich assemblage of geophysical wireline logs that include data from the BRG strata that occur at very shallow depths and behind casing [8]. This report utilizes 105 well logs (Fig 1; Table 1; S1–S4 Figs) selected during a review of thousands to update and refine recognition of formational contacts in the subsurface, and to document the extent and geometry of lithostratigraphic units and markers within the BRG. Fourteen reference well logs are tied to key outcrop sections (and fossiliferous locations) in southern Alberta and Saskatchewan (Fig 1) thus ensuring accurate calibration and correlation of stratigraphic cross-sections throughout the field area. Emphasis is placed on southeastern Alberta where very little subsurface work has been previously conducted and fossiliferous exposures of the BRG are abundant. Throughout this report, the following abbreviations are used for conciseness: BFm, Bearpaw Formation; BRG, Belly River Group; DPFm, Dinosaur Park Formation; FFm, Foremost Formation; Fm, Formation; IHS, inclined heterolithic strata; km, kilometers; m, meters; MRFm, Milk River Formation; MRS/MRs, Milk River shoulder; OFm, Oldman Formation; PFm, Pakowki Formation; Twp, Township; Rg, Range; S, Section; W, west of.

## Geologic context

The Belly River Group is an eastward-thinning paralic-to-nonmarine clastic wedge (Fig 2A) that was deposited as part of a 3$^{rd}$ order regressive-transgressive cycle in the Western Interior Basin (WIB) from ~81 Ma to ~74 Ma during the mid-to-late Campanian [15]. As redefined by Jerzykiewicz and Norris [2] and subsequently confirmed by numerous researchers [3–7], the BRG is bounded below and above by marine shales of the Pakowki/Lea Park (Claggett equivalent in the USA) and Bearpaw formations, respectively. The geographic limits of the BRG have been described previously [3–8] and are briefly reviewed here. The BRG can be recognized as far north as Edmonton and as far west as the deformed belt where Bearpaw shales pinch out. North of Edmonton, the BRG is equivalent to the lower Wapiti Formation. West of Calgary and into the deformed belt, the BRG is equivalent to the lower Brazeau Group. In the southwest foothills deformed area—beyond the range of reliable subsurface data and thus the limits

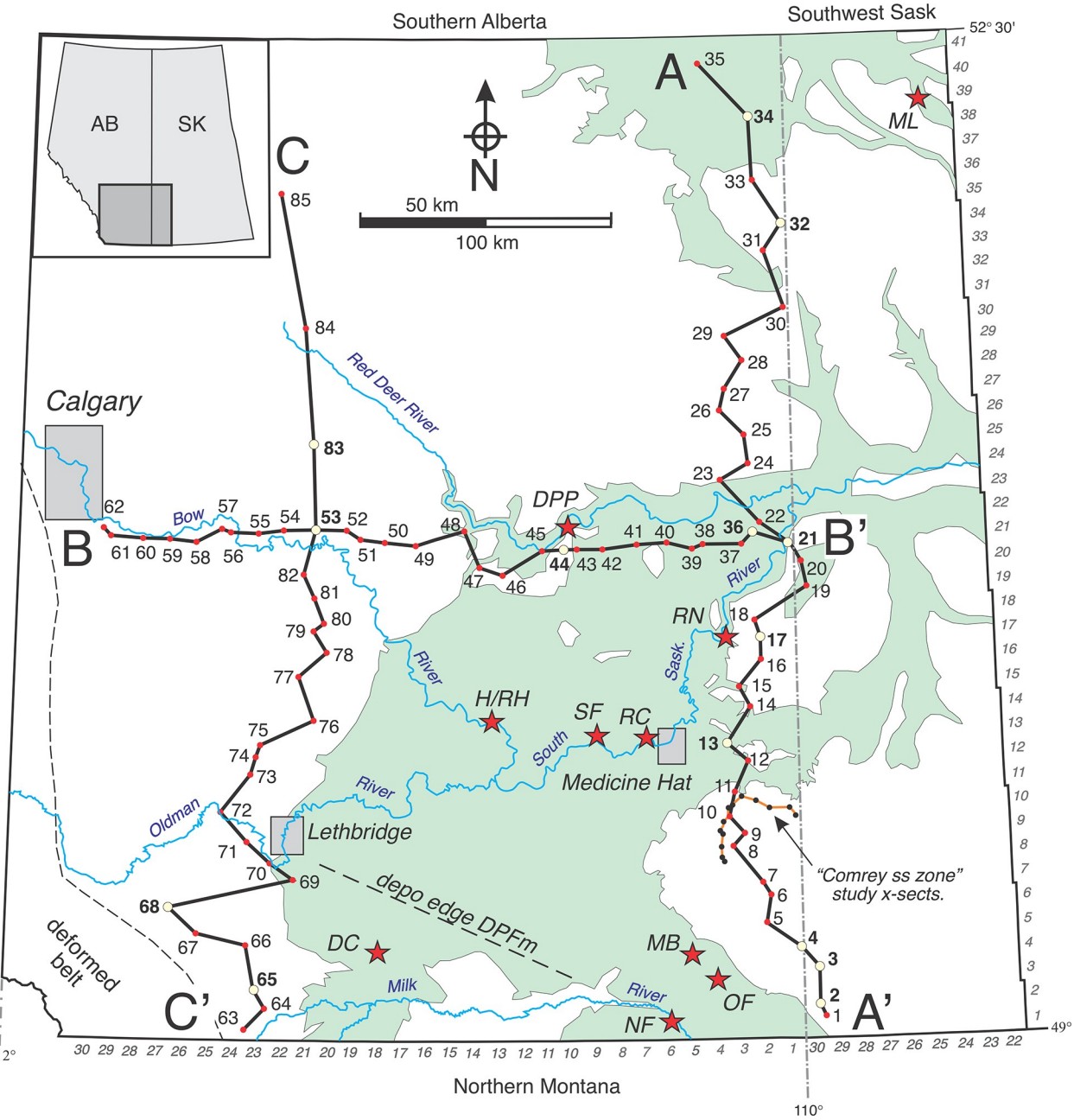

**Fig 1. Location map.** Light green indicates Belly River Group exposures in southern Alberta and southwest Saskatchewan. Inset shows field area in western Canada. Township and range numbers along margins. Cross-sections (A–A', B–B', C–C') indicated by solid lines. Well logs indicated by dots and numbers. Reference logs indicated by yellow dots and bold numbers. Outcrop locations indicated by red stars. Comrey sandstone zone study area and well log locations indicated southeast of Medicine Hat. Depositional edge of Dinosaur Park Fm indicated by dashed line southeast of Lethbridge. Abbreviations: DC, Devil's Coulee; DPP, Dinosaur Provincial Park; H/RH, Hays/Rolling Hills; OF, Onefour; MB, Manyberries; ML, Muddy Lake; NF, Nesmo Farm; RN, Rapid Narrows; RC, Redcliff; SF, Suffield Pumping Station; ss, sandstone.

of our study—the BRG has been divided into the Connelly Creek, Lundbreck, and Drywood Creek formations [2]. To the east in central Saskatchewan, the BRG thins stratigraphically and eventually pinches out in the marine shales of the Lea Park Formation. South of the international border, Belly River Group strata are equivalent to the Two Medicine and Judith River formations in northwestern and northcentral Montana, respectively [9].

**Table 1. Well log numbers, locations, and picks used in this study.**

| # | Ref | X-s | Location | MRs | FFm | OFm | DPFm | BFm |
|---|---|---|---|---|---|---|---|---|
| 1 | | AA' | 06-36-01-30W3 | 407 | 270 | 205 | 91 | ? |
| 2 | Ref | AA' | 14-14-02-30W3 | 412 | 272 | 217 | 91 | 54 |
| 3 | Ref | AA' | 08-36-03-30W3 | 531 | 384 | 324 | 205 | 167 |
| 4 | Ref | AA' | 11-25-04-01W4 | 519 | 369 | 316 | 193 | 155 |
| 5 | | AA' | 04-27-05-02W4 | 523 | 377 | 304 | 186 | 143 |
| 6 | | AA' | 05-35-06-02W4 | 671 | 539 | 456 | 352 | 315 |
| 7 | | AA' | 03-16-07-02W4 | 738 | 605 | 532 | 421 | 360 |
| 8 | | AA' | 04-32-08-03W4 | 657 | 533 | 456 | 344 | 278 |
| 9 | | AA' | 06-14-09-03W4 | 625 | 496 | 406 | 319 | 255 |
| 10 | | AA'Com | 06-06-10-03W4 | 564 | 424 | 353 | 256 | 195 |
| 11 | | AA' | 16-04-11-03W4 | 410 | 289 | 193 | 109 | 49 |
| 12 | | AA' | 09-13-12-03W4 | 337 | 210 | 114 | 55 | ? |
| 13 | Ref | AA' | 14-06-13-03W4 | 313 | 195 | 121 | 56 | ? |
| 14 | | AA' | 12-19-14-02W4 | 332 | 198 | 119 | 69 | ? |
| 15 | | AA' | 11-14-15-03W4 | 351 | 230 | 140 | 78 | 20 |
| 16 | | AA' | 14-22-16-02W4 | 424 | 317 | 227 | 156 | 94 |
| 17 | Ref | AA' | 16-15-17-02W4 | 406 | 271 | 200 | 140 | 77 |
| 18 | | AA' | 04-04-18-02W4 | 385 | 282 | 188 | 135 | 83 |
| 19 | | AA' | 12-24-19-29W3 | 373 | 265 | 180 | 132 | 81 |
| 20 | | AA' | 16-04-20-29W3 | 356 | 233 | 156 | 109 | 65 |
| 21 | Ref | AA'BB' | 06-12-21-01W4 | 361 | 235 | 165 | 106 | 60 |
| 22 | | AA' | 03-02-22-02W4 | 328 | 238 | 131 | 76 | ? |
| 23 | | AA' | 14-31-23-03W4 | 359 | 244 | 153 | 103 | 55 |
| 24 | | AA' | 12-20-24-02W4 | 321 | 208 | 110 | 81 | 23? |
| 25 | | AA' | 16-30-25-02W4 | 372 | 262 | 171 | 139 | 90 |
| 26 | | AA' | 11-30-26-03W4 | 372 | 257 | 171 | 138 | 78 |
| 27 | | AA' | 15-21-27-03W4 | 404 | 288 | 192 | 146 | 78 |
| 28 | | AA' | 15-30-28-02W4 | 375 | 260 | 164 | 134 | 75 |
| 29 | | AA' | 08-28-29-03W4 | 362 | 220 | 145 | 110 | 48? |
| 30 | | AA' | 16-36-30-01W4 | 337 | 217 | 113 | 82 | 29 |
| 31 | | AA' | 07-07-33-01W4 | 372 | 234 | 149 | 113 | 55 |
| 32 | Ref | AA' | 13-13-34-01W4 | 396 | 255 | 173 | 149 | 96 |
| 33 | | AA' | 16-01-36-02W4 | 449 | 309 | 220 | 195 | 149 |
| 34 | Ref | AA' | 03-27-38-02W4 | 501 | 328 | 275 | 262 | 200 |
| 35 | | AA' | 10-34-40-04W4 | 371 | 224 | 145 | 134 | 70 |
| 36 | Ref | BB' | 07-28-21-02W4 | 320 | 217 | 125 | 67 | ? |
| 37 | | BB' | 14-12-21-03W4 | 312 | 217? | 128 | 70 | ? |
| 38 | | BB' | 08-08-21-04W4 | 309 | 220 | 119 | 69 | ? |
| 39 | | BB' | 14-01-21-05W4 | 355 | 288 | 156 | 91 | ? |
| 40 | | BB' | 06-13-21-06W4 | 345 | 274 | 146 | 88 | ? |
| 41 | | BB' | 15-10-21-07W4 | 375 | 304 | 175 | 125 | 44 |
| 42 | | BB' | 02-05-21-08W4 | 340 | 284 | 135 | 92 | ? |
| 43 | | BB' | 14-04-21-09W4 | 337 | 285 | 129 | 84 | ? |
| 44 | Ref | BB' | 04-02-21-10W4 | 378 | 323 | 169 | 125 | 39 |
| 45 | | BB' | 05-05-20-11W4 | 326 | 272 | 116 | 85 | ? |
| 46 | | BB' | 08-17-20-12W4 | 338 | 296 | 134 | 100 | 13 |
| 47 | | BB' | 01-34-21-13W4 | 324 | 267 | 118 | 73 | ? |

*(Continued)*

**Table 1.** (Continued)

| # | Ref | X-s | Location | MRs | FFm | OFm | DPFm | BFm |
|---|---|---|---|---|---|---|---|---|
| 48 | | BB' | 05-22-21-14W4 | ? | 372 | 155 | 112 | 21 |
| 49 | | BB' | 15-10-21-15W4 | 389 | 331 | 181 | 133 | 59 |
| 50 | | BB' | 10-16-21-16W4 | 398 | 332 | 185 | 136 | 56 |
| 51 | | BB' | 02-21-21-17W4 | ? | 425 | 208 | 160 | 83 |
| 52 | | BB' | 08-35-21-18W4 | ? | 492 | 263 | 213 | 128 |
| 53 | Ref | **BB'CC'** | 08-34-21-20W4 | 593 | 534 | 366 | 329 | 244 |
| 54 | | BB' | 11-32-21-21W4 | 666 | 618 | 431 | 393 | 314 |
| 55 | | BB' | 14-29-21-22W4 | 737 | 692 | 503 | 462 | 380 |
| 56 | | BB' | 14-30-21-23W4 | 779 | 732 | 547 | 500 | 416 |
| 57 | | BB' | 14-35-21-24W4 | 809 | 785 | 570 | 530 | 446 |
| 58 | | BB' | 03-15-21-25W4 | 913 | 879 | 674 | 631 | 537 |
| 59 | | BB' | 06-22-21-26W4 | 983 | 935 | 735 | 701 | 608 |
| 60 | | BB' | 11-02-21-27W4 | ? | 1177 | 939 | 907 | 797 |
| 61 | | BB' | 03-20-21-28W4 | 1225 | 1199 | 973 | 946 | 841 |
| 62 | | BB' | 11-36-21-29W4 | 1214 | 1185 | 961 | 921 | 820 |
| 63 | | CC' | 12-18-01-23W4 | ? | 1097 | 863 | NA | 641 |
| 64 | | CC' | 01-13-02-23W4 | 994 | 986 | 743 | NA | 542 |
| 65 | Ref | CC' | 01-11-03-23W4 | 861 | 855 | 629 | NA | 429 |
| 66 | | CC' | 03-32-04-23W4 | 805 | 788 | 578 | NA | 381 |
| 67 | | CC' | 09-17-05-25W4 | 1274 | 1260 | 1068 | NA | 830 |
| 68 | Ref | CC' | 16-18-06-26W4 | 1669 | 1646 | 1483 | NA | 1208 |
| 69 | | CC' | 10-29-07-21W4 | 383 | 356 | 214 | NA | ? |
| 70 | | CC' | 15-08-08-22W4 | 630 | 585 | 406 | NA | 174 |
| 71 | | CC' | 02-04-09-23W4 | 815 | 774 | 617 | 438 | 399 |
| 72 | | CC' | 13-16-10-24W4 | 417 | 390 | 217 | 60 | 30 |
| 73 | | CC' | 05-34-11-23W4 | 709 | 683 | 523 | 419 | 348 |
| 74 | | CC' | 03-23-12-23W4 | ? | 695 | 528 | 445 | 366 |
| 75 | | CC' | 04-06-13-22W4 | 652 | 618 | 467 | 374 | 298 |
| 76 | | CC' | 06-05-14-20W4 | 449 | 418 | 259 | 180 | 97 |
| 77 | | CC' | 01-35-15-21W4 | 591 | 552 | 408 | 290 | 244 |
| 78 | | CC' | 14-36-16-20W4 | 500 | 466 | 283 | 210 | 155 |
| 79 | | CC' | 06-28-17-20W4 | 625 | 584 | 417 | 349 | 275 |
| 80 | | CC' | 06-02-18-20W4 | 592 | 551 | 360 | 288 | 225 |
| 81 | | CC' | 14-03-19-20W4 | 644 | 600 | 419 | 351 | 278 |
| 82 | | CC' | 04-06-20-20W4 | 646 | 600 | 417 | 365 | 296 |
| 83 | Ref | CC' | 15-15-25-20W4 | 707 | 650 | 480 | 455 | 370 |
| 84 | | CC' | 09-04-30-20W4 | 926 | 888 | 698 | 675 | 566 |
| 85 | | CC' | 15-27-35-21W4 | 682 | 682 | 452 | 448 | 326 |
| 86 | | Com | 16-20-10-03W4 | 475 | 355 | 260 | 165 | 111 |
| 87 | | Com | 06-34-10-03W4 | 439 | 315 | 230 | 130 | 70 |
| 88 | | Com | 07-14-10-02W4 | 528 | 409 | 311 | 215 | 155 |
| 89 | | Com | 11-29-10-02W4 | 510 | 401 | 305 | 221 | 145 |
| 90 | | Com | 06-36-10-02W4 | 497 | 381 | 292 | 200 | 136 |
| 91 | | Com | 06-06-10-01W4 | 579 | 454 | 363 | 289 | 205 |
| 92 | | Com | 11-29-10-01W4 | 475 | 352 | 257 | 168 | 112 |
| 93 | | Com | 06-22-10-01W4 | 463 | 343 | 248 | 153 | 97 |
| 94 | | Com | 10-11-10-01W4 | 485 | 342 | 275 | 191 | 122 |

(*Continued*)

**Table 1.** (Continued)

| # | Ref | X-s | Location | MRs | FFm | OFm | DPFm | BFm |
|---|-----|-----|----------|-----|-----|-----|------|-----|
| 95 | | Com | 06-02-10-01W4 | 504 | 359 | 298 | 202 | 140 |
| 96 | | Com | 06-18-10-03W4 | 500 | 380 | 296 | 184 | 131 |
| 97 | | Com | 06-36-09-04W4 | 567 | 451 | 366 | 254 | 195 |
| 98 | | Com | 13-25-09-04W4 | 504 | 436 | 349 | 240 | 177 |
| 99 | | Com | 01-23-09-04W4 | 551 | 427 | 354 | 241 | 174 |
| 100 | | Com | 04-14-09-04W4 | 666 | 540 | 460 | 340 | 291 |
| 101 | | Com | 06-02-09-04W4 | 621 | 500 | 410 | 297 | 243 |
| 102 | | Com | 15-35-08-04W4 | 573 | 449 | 370 | 258 | 195 |
| 103 | | Com | 09-26-08-04W4 | 593 | 467 | 374 | 290 | 210 |
| 104 | | Com | 11-23-08-04W4 | 567 | 442 | 350 | 263 | 212 |
| 105 | | Com | 16-12-08-04W4 | 790 | 569 | 473 | 384 | 307 |

Notes

All measurements in meters

Formation picks are bottoms

Abbreviations: ref, reference logs; x-s, cross-section; Com, comrey ss study area; na, not applicable.

Other abbreviations as per text.

Throughout most of the plains, and in ascending order, the BRG consists of the Foremost, Oldman, and Dinosaur Park formations (Fig 2A) [1, 3–8]. The three formations record significant adjustments in provenance, sediment supply and distributional directions over millions of years (Fig 2B), with the Oldman Fm representing a wedge of sediment sourced largely from volcanic terranes in northwestern Montana, and the Foremost and Dinosaur Park formations sourced farther north from centers in the Cordillera of central British Columbia [1, 6, 7, 16].

The Foremost Fm is a coaly, marine-to-paralic unit that prograded eastward and marked the retreat from Alberta of marine environments (Pakowki/Lea Park Fm) from ~80–78 Ma [8]. The Oldman Fm is a predominantly nonmarine alluvial unit that records the maximum retreat of the WIS from Alberta and western Saskatchewan at ~78–77 Ma [1, 7, 11, 12]. It thickens considerably in southwestern Alberta where it replaces the Dinosaur Park Fm and becomes younger up-section. To the north, it is eventually replaced by the Foremost Fm. As the Oldman Fm loses distinction to the north (~Twp 40), the Foremost and Dinosaur Park formations amalgamate, but are difficult to differentiate. The Dinosaur Park Fm is a marine-paralic and nonmarine alluvial unit that records the transgression of the Bearpaw Sea across western Saskatchewan and southern Alberta from ~77–74 Ma [1, 7, 9, 11, 12, 14]. It pinches out stratigraphically to the southwest in the Lethbridge area and thins toward the southeast (Manyberries area) where it extends into northern Montana. There, it may be the stratigraphic equivalent of the Coal Ridge Member of the Judith River Fm [1, 14].

Discussions of the complex nomenclatural history of the Belly River Group and its lithostratigraphic subdivisions in Canada are available in a variety of sources [1, 3, 7–9, 17, 18] and are not presented or reviewed here.

## Methods

Raster well-log access and use was graciously provided by Divestco Geoscience Ltd. (Calgary, Alberta) through their "Energisite" website and platform. All well logs employed herein are listed in Table 1, and are presented here in annotated form in Figs 3–8, 12–14, and S1–S4 Figs. Well log data in Table 1 and illustrated well logs include formational picks presented as depth-

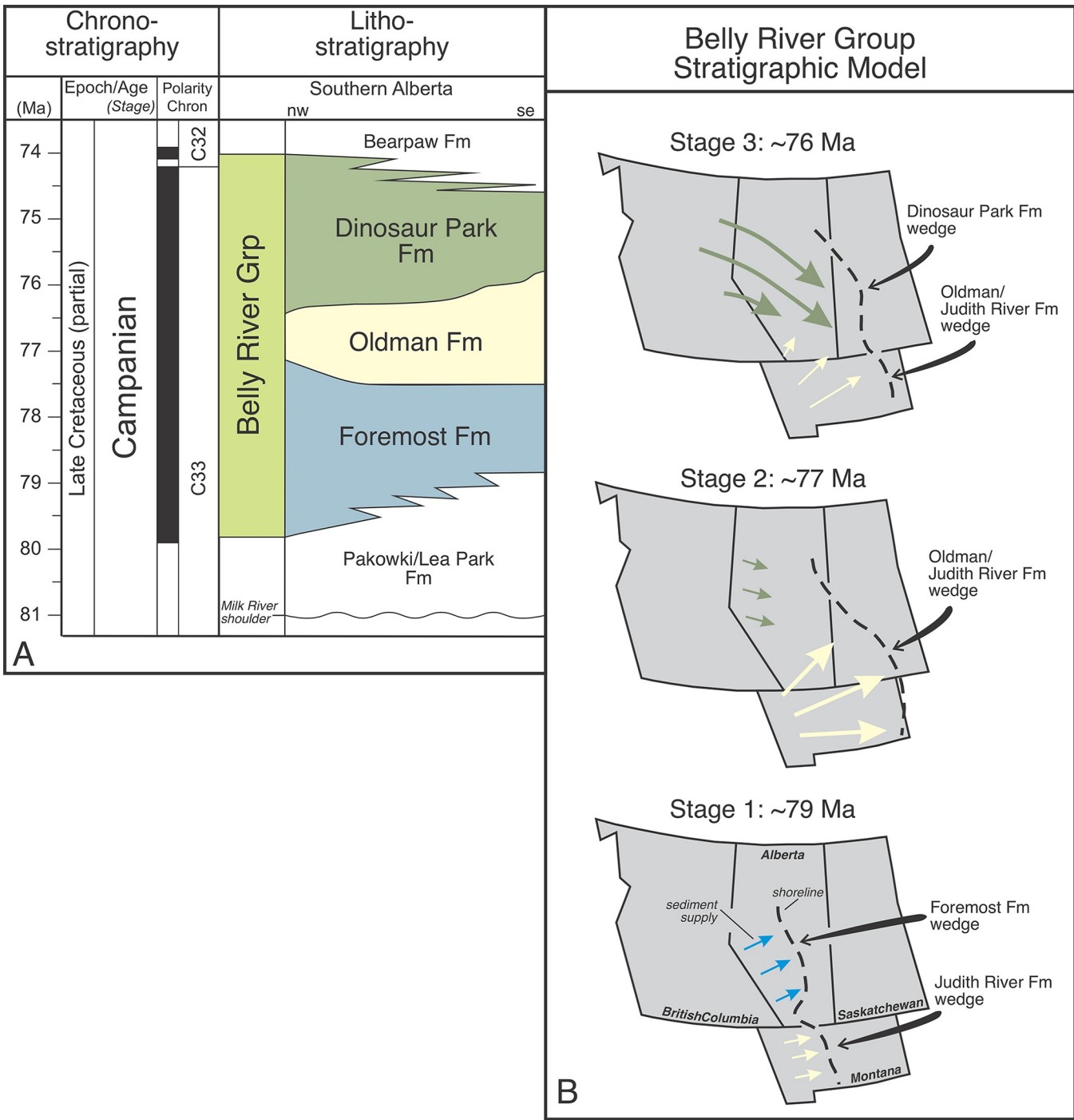

**Fig 2. Stratigraphy and stratigraphic model of the Belly River Group.** A, Stratigraphic chart for the Belly River Group. Note the complex geometries of the Foremost, Oldman, and Dinosaur Park formations, and the step-like nature of the Dinosaur Park-Oldman formational contact in southeastern Alberta. B, Stratigraphic model for the Belly River Group showing shifting sediment supply directions and intensities through time. Abbreviations: Ma, Mega-annum; nw, northwest; se, southeast.

from-surface measurements in meters. Gamma-ray and "porosity" logs (density, neutron, and sonic) are required to accurately differentiate the Foremost, Oldman, and Dinosaur Park formations in the Belly River Group [1, 4–6, 8, 9], and gamma-density (or neutron) pairs were

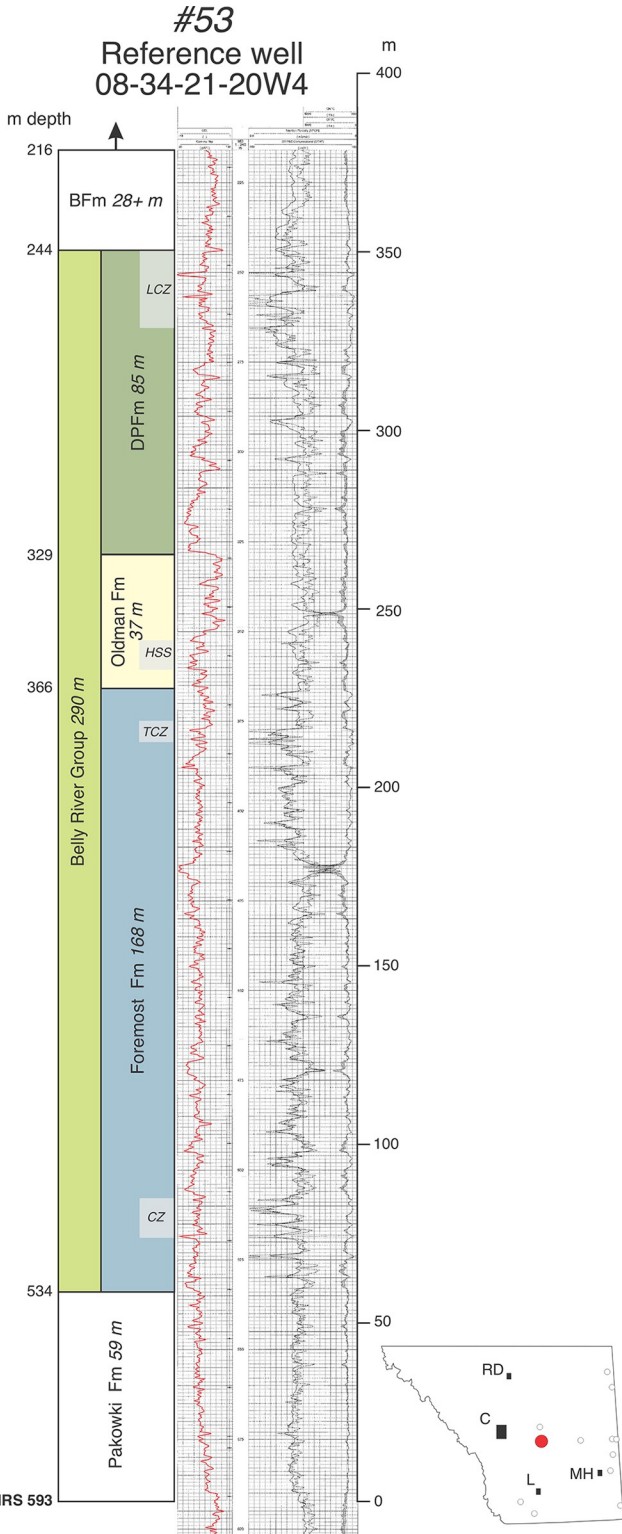

**Fig 3. Annotated reference well log #53 (08-34-21-20W4) from center of the field area (between Bassano and Gleichen) for comparison with Hamblin [4, his Fig 3].** Inset shows location (red dot) in southern Alberta relative to other reference logs (circles). Gamma (red) and neutron-porosity log patterns are typical of the Belly River Group north of Twp 19 in southern Alberta. Note relative thinness of the Oldman Formation compared to more southerly logs (Figs 4 and 5). Abbreviations: LCZ, Lethbridge coal zone; HSS, Herronton sandstone zone; CZ, coal zone; TCZ,

Taber coal zone; MRS, Milk River shoulder. Formational abbreviations as in text. Abbreviations in inset indicate locations of Calgary, Lethbridge, Medicine Hat, and Red Deer.

mostly utilized here. Well log locations are reported in Table 1 and log headings using the standard legal land location designation for Alberta (Legal Subdivision-Section-Township-Range). Unless otherwise indicated (e.g., wells and outcrop sections in southwestern Saskatchewan), locations occur in Alberta west of the 4th Meridian.

Most well logs used here employ the Milk River "shoulder" as a datum for stratigraphic correlation and the compilation of cross-sections. The shoulder reflects a widespread unconformity and a sharp up-section transition from coarser (mudstones and sandstones) to finer sediments (shales) at the base of the Pakowki Fm (in southern Alberta) and the top of the Alderson Member within the Lea Park Fm at more distal locations to the north and east [4, 19]. The MRs can be traced to the southeast into Montana [9] but loses distinction in southwestern Alberta, near the international border, due to thinning and stratigraphic pinch-out of the marine Pakowki Fm. In that area, the top of the Milk River Fm is used as a datum.

Fourteen reference logs (Figs 3–8) were utilized here based on their clarity and effectiveness in characterizing typical well-log signatures for the stratigraphic units discussed here in different geographic regions of southern Alberta. Three subsurface cross-sections were compiled covering the full range of expression of the BRG in southern Alberta (Fig 1). A one-well-per-township spacing was assessed as sufficient to illustrate the formation-scale stratigraphic patterns described here. In areas north of Twp 30, where the Oldman Fm thins to less than 20 m and Belly River stratigraphy becomes more monotonous, well-log spacing was increased to 2–5 townships or more. In contrast, a one-well-per-section spacing was employed to test hypotheses regarding the continuity of the "Comrey Sandstone" and "Comrey sandstone zone" in southeastern Alberta (Fig 1). Subsurface data from the deformed belt, which is characterized by structurally complex strata with repeated stratigraphic intervals and a different stratigraphic nomenclature [2], are not included in this study.

Ten measured outcrop locations are indicated in Fig 1 and listed in Table 2. They include vertebrate-fossil-rich strata reported in the literature and typically include at least one formational contact to facilitate correlation with geophysical data from nearby wells. Sections were measured using a 1.5 m Jacob's staff with a top-mounted Abney level. As presented here, section graphics depict grain size and basic sedimentary structures. Colors in the graphics reflect increasing amounts of organic content in fine grained deposits (yellows-to-browns), iron-staining in sandstones (red-to-purple), and the presence of airfall bentonites (green).

**Table 2. List of measured section names and locations used in this study.**

| Name | Location | Drainage | Dates measured | Formations present |
|---|---|---|---|---|
| Nesmo Farm | 01-06W4 | Milk River east | 10/25/88; 07/08/91 | FFm, OFm |
| Onefour East | 03-04W4 | Milk River east | 06/14/89 | OFm, DPFm, BFm |
| S. Manyberries Crk | 35-04-05W4 | Milk River east | 06/15/89 | OFm, DPFm, BFm |
| Devil's Coulee | 04-18W4 | Milk River Ridge | 05/31/89; 06/01-03/89 | middle OFm |
| Redcliff | 05-13-06W4 | S. Sask. River | 06/17/87 | FFm, OFm |
| Suffield Pump | 11-13-09W4 | S. Sask. River | 06/18/93 | PFm, FFm, OFm |
| Hays/Rolling Hills | 07-02-14-13W4 | Bow River | after Hamblin (1997a) | FFm, OFm |
| Rapid Narrows | 07-17-03W4 | S. Sask. River | 07/18/89 | FFm, OFm, DPFm |
| Dinosaur Prov. Park | 01-22-10W4 | Red Deer River | 07/23-24/96 | OFm, DPFm, BFm |
| Muddy Lake, SK | 39-23W3 | Muddy Lake | 06/16/88; 09/19/89 | FFm, OFm, DPFm |

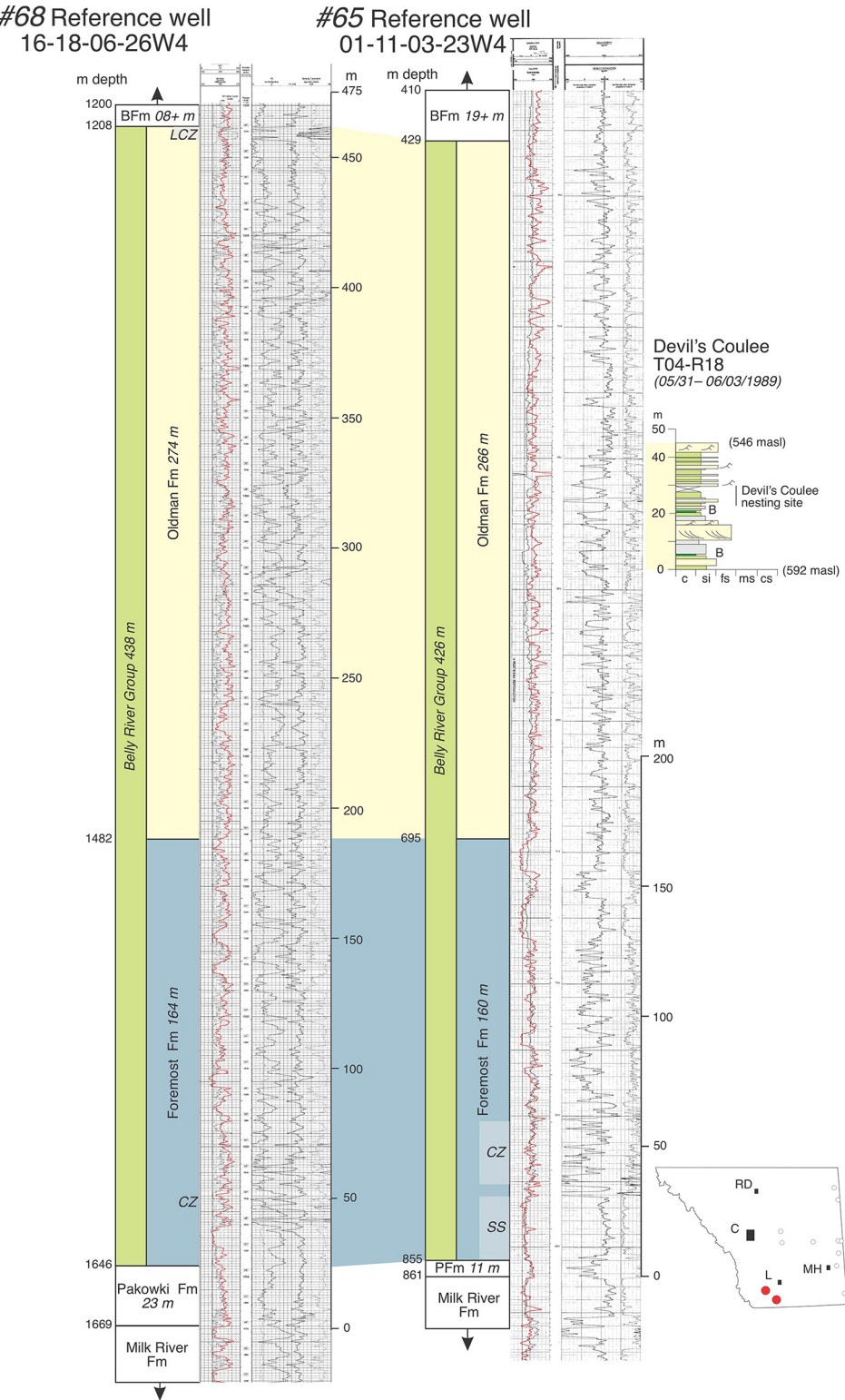

**Fig 4. Annotated reference logs #65 (01-11-03-23W4) and #68 (16-18-06-26W4) from southwestern corner of field area.** Note extreme thickness of the Oldman Fm in this area. Also note the measured section of fossiliferous strata from Devil's Coulee, west of Warner (see Fig 1). Measured section was stratigraphically correlated to well log #68 using field observations of DAE. Abbreviations: B, datable bentonite; c, claystone; cs, coarse sandstone; fs, fine sandstone; masl, meters above sea level; ms, medium sandstone; PFm, Pakowki Formation; si, siltstone.

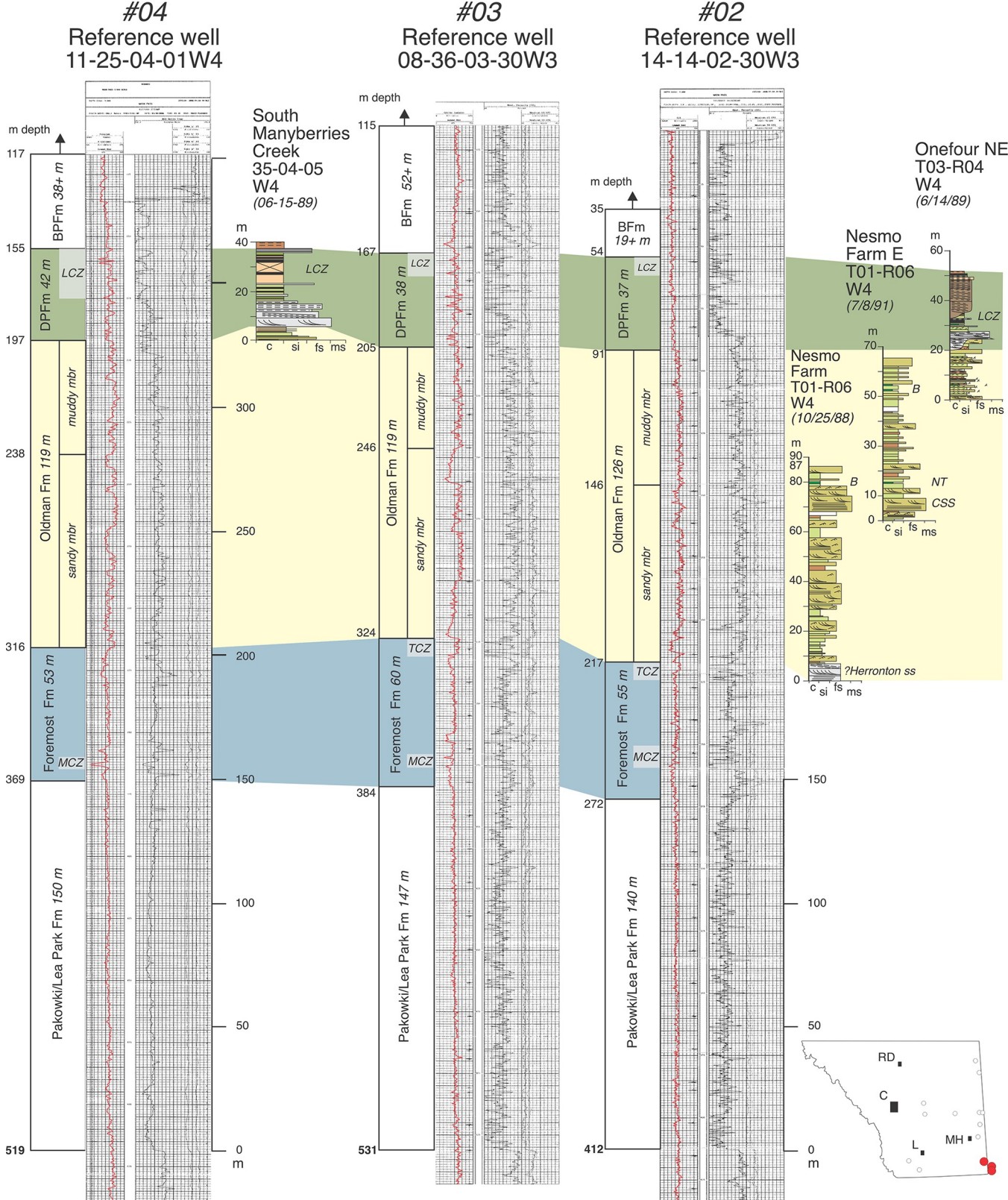

**Fig 5. Annotated reference logs #02 (14-14-02-30W3), #03 (08-36-03-30W3), and #04 (11-25-04-01W4) from southeast Alberta.** Note the thickness of the Oldman Fm. Also included are measured sections of fossiliferous strata from Onefour, Nesmo Farm, and South Manyberries Creek (southeast of Manyberries).

All sections are stratigraphically correlated to one another and well logs using local marker beds and formational contacts observed by DAE. Abbreviations: CSS, Comrey Sandstone; LCZ, Lethbridge coal zone; MCZ, McKay coal zone; NT, Nesmo tuff.

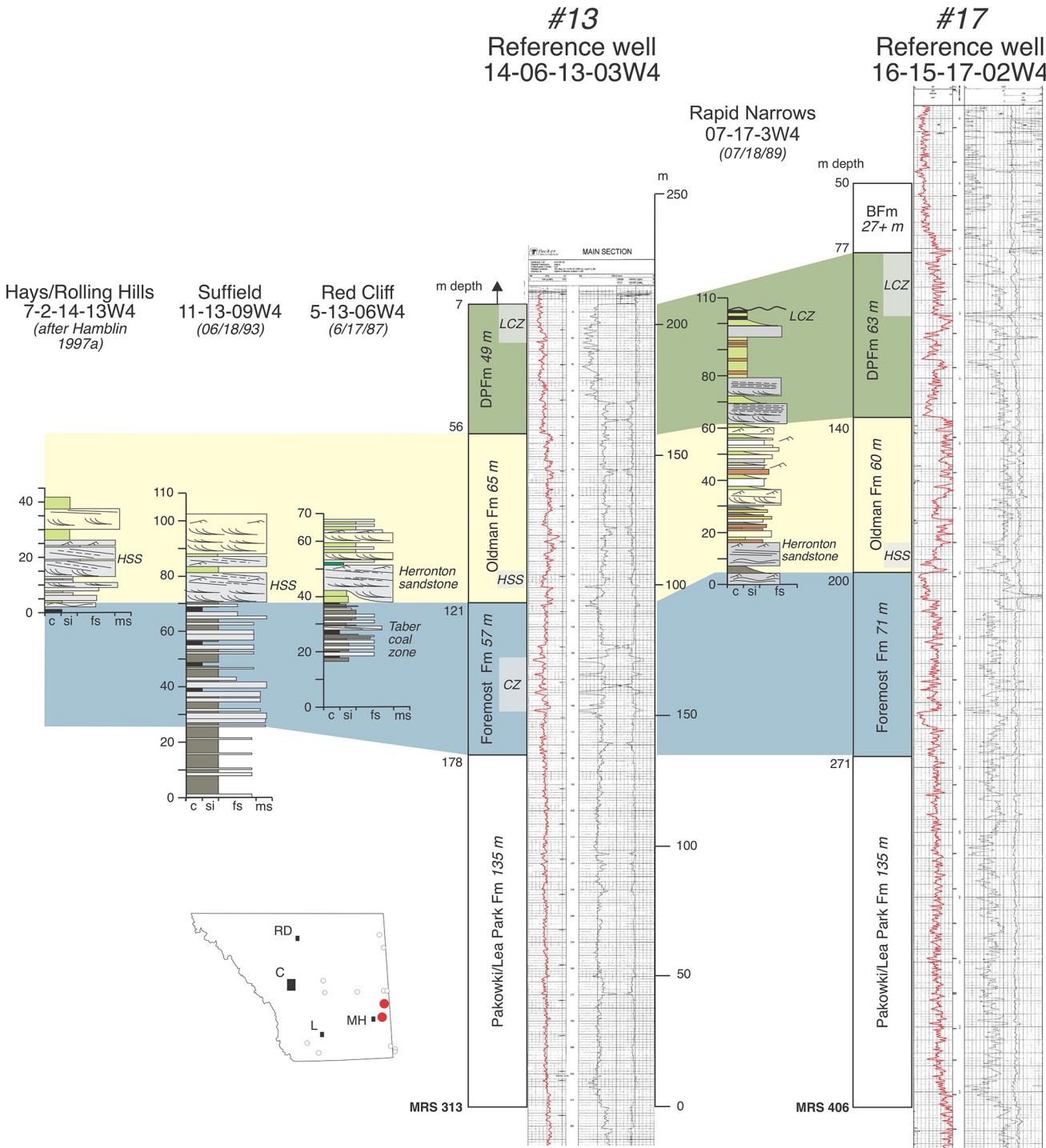

**Fig 6. Annotated reference logs #13 (14-06-13-03W4) and #17 (16-15-17-02W4) from Medicine Hat area of southeast Alberta.** Also included are measured sections of fossiliferous strata from the Bow and South Saskatchewan rivers near Medicine Hat. All sections are correlated to one another and well logs using local marker beds, formational contacts, and formation thicknesses observed by DAE as well as Hamblin [4]. Note the Herronton sandstone at the base of the Oldman in the measured sections.

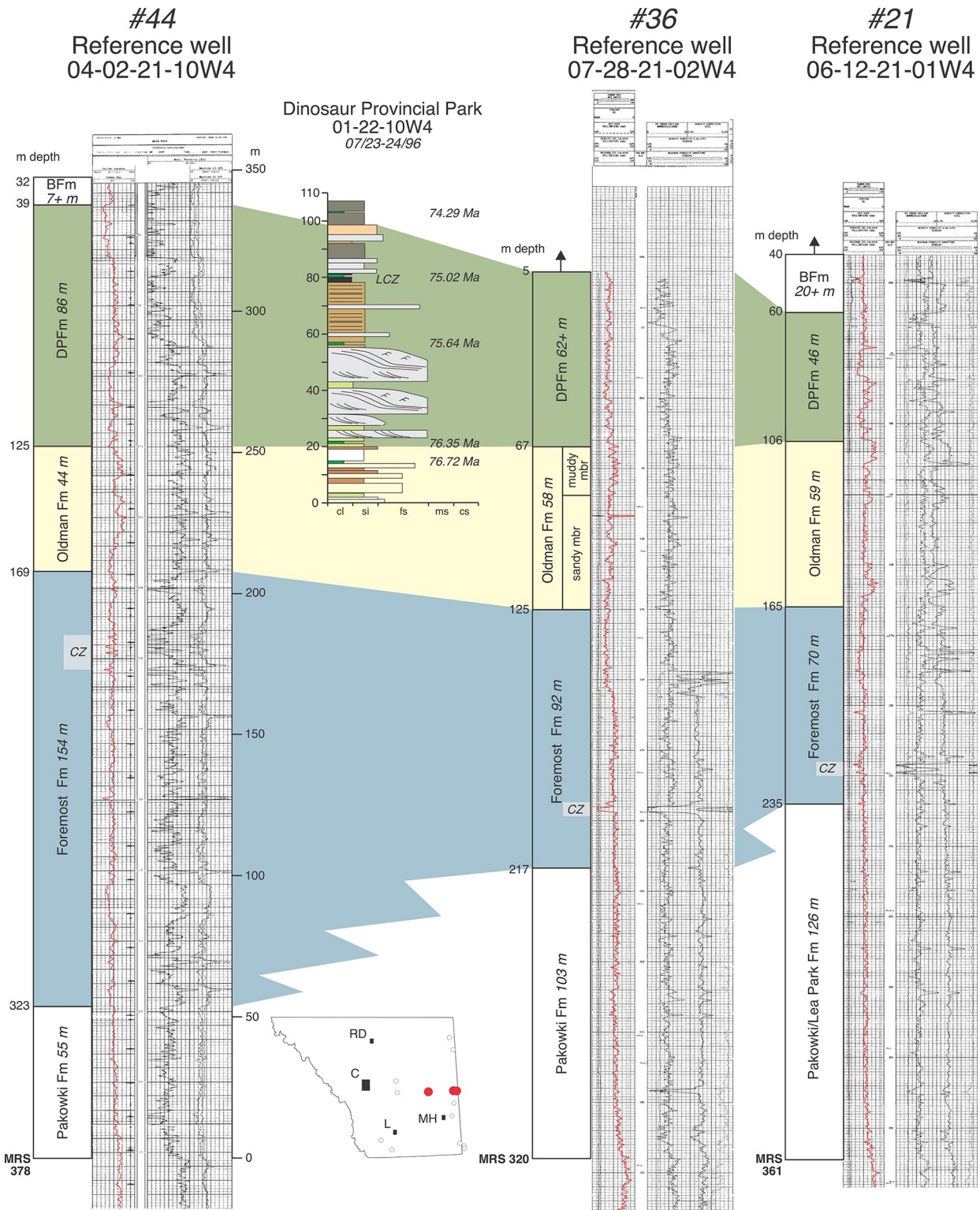

**Fig 7. Annotated reference logs #21 (06-12-21-01W4), #36 (07-28-21-02W4), and #44 (04-02-21-10W4) from Dinosaur Provincial Park and areas east.** Also included is a measured section of fossiliferous strata from the Red Deer River at Dinosaur Provincial Park. All sections are correlated to one another and well logs using local marker beds, formational contacts, and formation thicknesses [1, 7, 11, 12]. The five U-Pb radiometric ages from the Oldman, Dinosaur Park and Bearpaw formations at Dinosaur Provincial Park are from Ramezani et al. [11] and Eberth et al. [12]. Also note the sandy lower one-half and muddy upper one-half of the Dinosaur Park Fm.

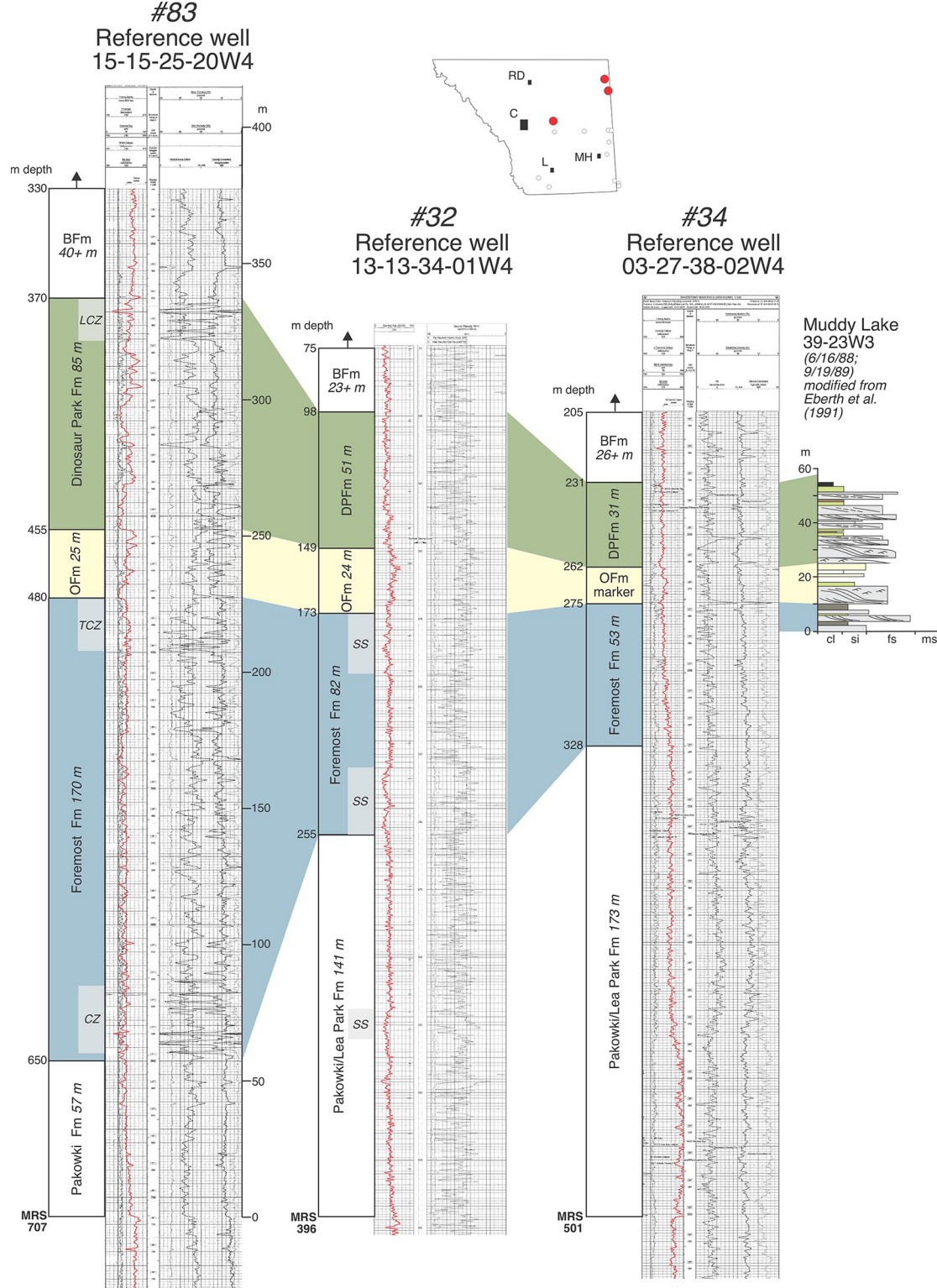

**Fig 8. Annotated reference logs #32 (13-13-34-01W4), #34 (03-27-38-02W4), and #83 (15-15-25-20W4) from the northern part of the field area.** Also included is a measured section from Muddy Lake, Saskatchewan [20]. Note the westward stratigraphic thickening of the Foremost and Dinosaur Park formations relative to the Oldman Fm.

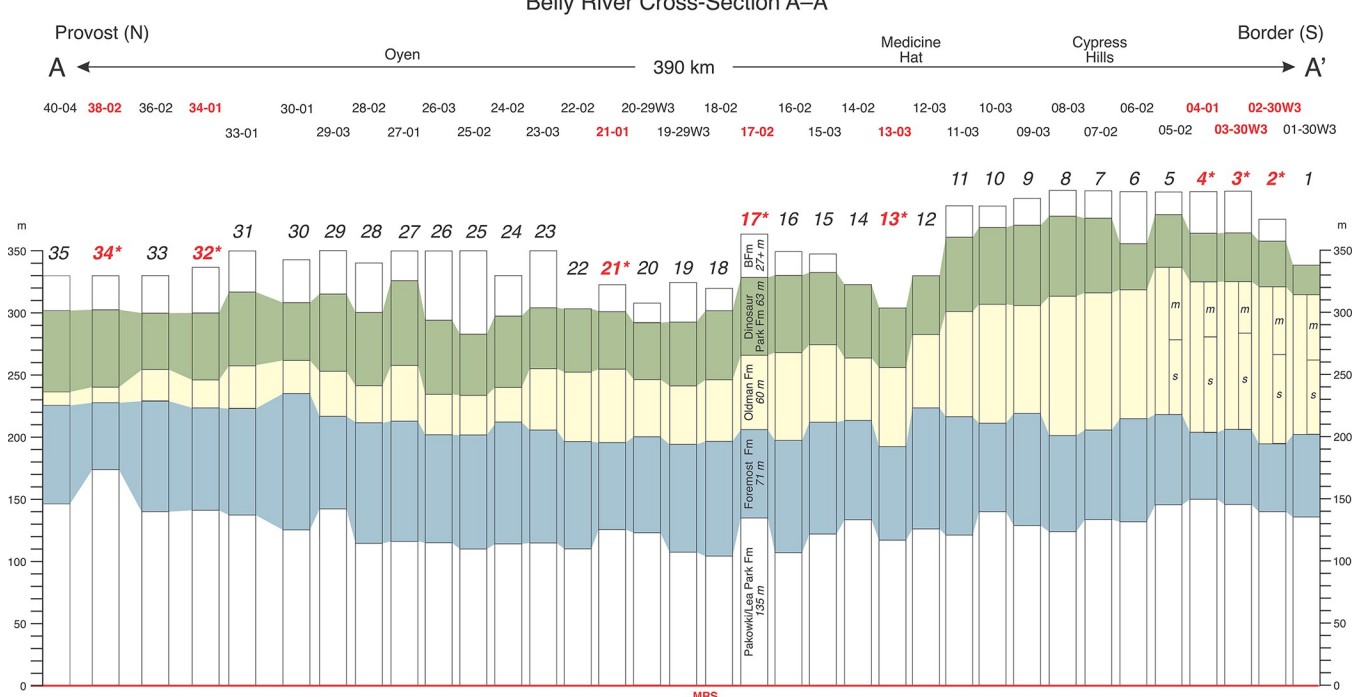

**Fig 9. Stratigraphic cross-section A–A'.** Cross-section A-A' is based on data from 35 wells logs (Fig 1, Table 1, S1 Fig). Logs are not illustrated in the cross-section due to image-size constraints. Instead, the cross-section exhibits color-coded formational thicknesses within and between adjacent wells. Numbers relate to well locations shown in Fig 1. Numbers in red indicate reference logs shown in detail in Figs 5–8. Note the southward thickening (mostly up-section) of the Oldman Formation and the commensurate thinning of the Dinosaur Park Fm in southeastern-most, as originally hypothesized by Eberth and Hamblin [1]. The Foremost-Oldman formational contact rises slightly to the north due to stratigraphic thinning of the Oldman Fm. Also note the occurrence of a well-developed two-fold stratigraphic division of the Oldman Fm in the southern-most corner of the field area (Twp 01–05). This pattern of a lower sandstone dominated interval overlain by a mudstone dominated interval compares closely to that described by Rogers et al. [9, 14] for the Judith River Formation. The contact between the two intervals in southeast Alberta thus likely marks the mid-Judith discontinuity (MJD) described by those authors.

## Characterizing the Belly River Group in well logs

The lower and upper boundaries of the BRG are marked by interfingering and thus diachronous transitions between marine and paralic facies that make precise placement of formational contacts somewhat arbitrary in any given location or well [17, 18, 21, 22]. For consistency and convenience in this report, the lower contact of the Foremost Fm on the Pakowki/Lea Park Fm marine shales is placed at the base of the first prominent upward-coarsening sandstone that also marks the base of a succession of interbedded shales, coals, siltstones, and sandstones that is ultimately overlain by the non-marine Oldman Fm (e.g., Fig 3). In this context, marine shales may be included within the formation but do not dominate. Exceptions to this pattern occur in southwestern Alberta near the international border. There, the Pakowki Fm thins and carbonaceous, paralic strata of the lower Foremost Fm approach or amalgamate with sandy paralic strata of the Milk River Fm (e.g., Fig 4). Likewise, the upper boundary of the BRG, which consists of the contact of Bearpaw Fm (marine shales) on the Dinosaur Park Fm (paralic sediments) is placed above the highest sandstones, siltstones, coals, and carbonaceous shales of the Dinosaur Park Fm, at a position where shales begin to dominate the overlying succession (Fig 3). Exceptions to this pattern occur west of Calgary where marine shales of the Bearpaw Fm pinch out and the Belly River Group transitions laterally into the lower portion of the Campanian-Maastrichtian age non-marine Brazeau Group.

Whereas picking the lower and upper bounding contacts of the BRG in the subsurface is a relatively straightforward process, picking the FFm-OFm and OFm-DPFm contacts requires

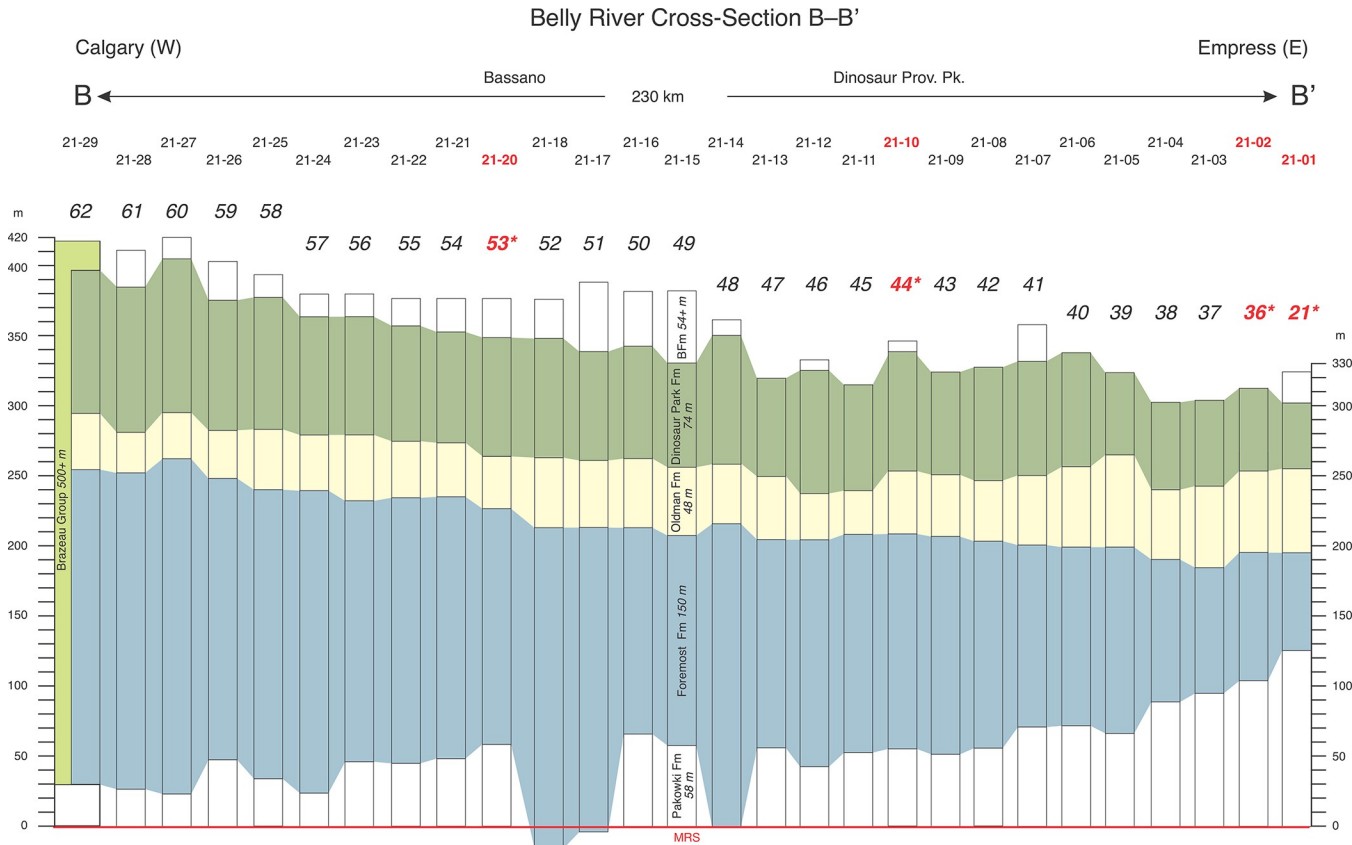

**Fig 10. Stratigraphic cross-section B–B'.** Cross-section B-B' is based on data from 28 well logs (Fig 1, Table 1, S2 Fig) and shows the mostly tabular geometry of the Oldman Fm that is maintained W–E (proximal-distal) at Twp 21. In contrast, both the Foremost and Dinosaur Park formations exhibit stratigraphic thickening to the west (toward source). Thus, the mostly tabular W-E geometry of the Oldman Fm confirms its separate and more southwesterly source area. Note also that west of Rg 28 (Calgary) the Bearpaw Fm pinches out and the Belly River Group is equivalent to the lower portion of the much thicker non-marine Brazeau Group.

greater consideration. Dowling's [23] two-fold division of the BRG (Foremost and Pale beds) was further refined by Russell and Landes [17] who elevated those units to formational rank and renamed the Pale beds the Oldman Fm (based on "type" exposures along the Oldman River between Lethbridge and the downstream confluence with the Bow River). Since that time, the Foremost-Oldman formational contact in the southern Alberta plains has been placed in the middle of the BRG, at the top of the upper-most coal/carbonaceous shale succession in the Foremost Fm and/or at the base of the first fluvial sandstone in the overlying silty-sandy, non-coaly alluvial succession (Oldman Fm; e.g., [1, 4–6, 21]).

McLean [18] emphasized electric logs in his subsurface study of the Belly River Group (his Judith River Fm) but could not differentiate with any consistency the FFm-OFm formational contact in either outcrop or subsurface as described by previous authors. In contrast, Eberth and Hamblin [1] and subsequent workers [3–9] have emphasized gamma-ray logs in their study of the BRG and have recognized a consistent and significantly high gamma-ray response through muddy portions of the Oldman Fm (Fig 3; also visible in the subsurface cross-sections of MacDonald et al. [21]. Eberth and Hamblin [1] studied the region surrounding Dinosaur Provincial Park (Twp 20-Rg 12) in detail, formally recognized the uppermost ~80 m of the Oldman Fm as a new unit, the Dinosaur Park Fm, and used the relatively high gamma-ray response in the remaining Oldman Fm as a means to identify and differentiate that formation

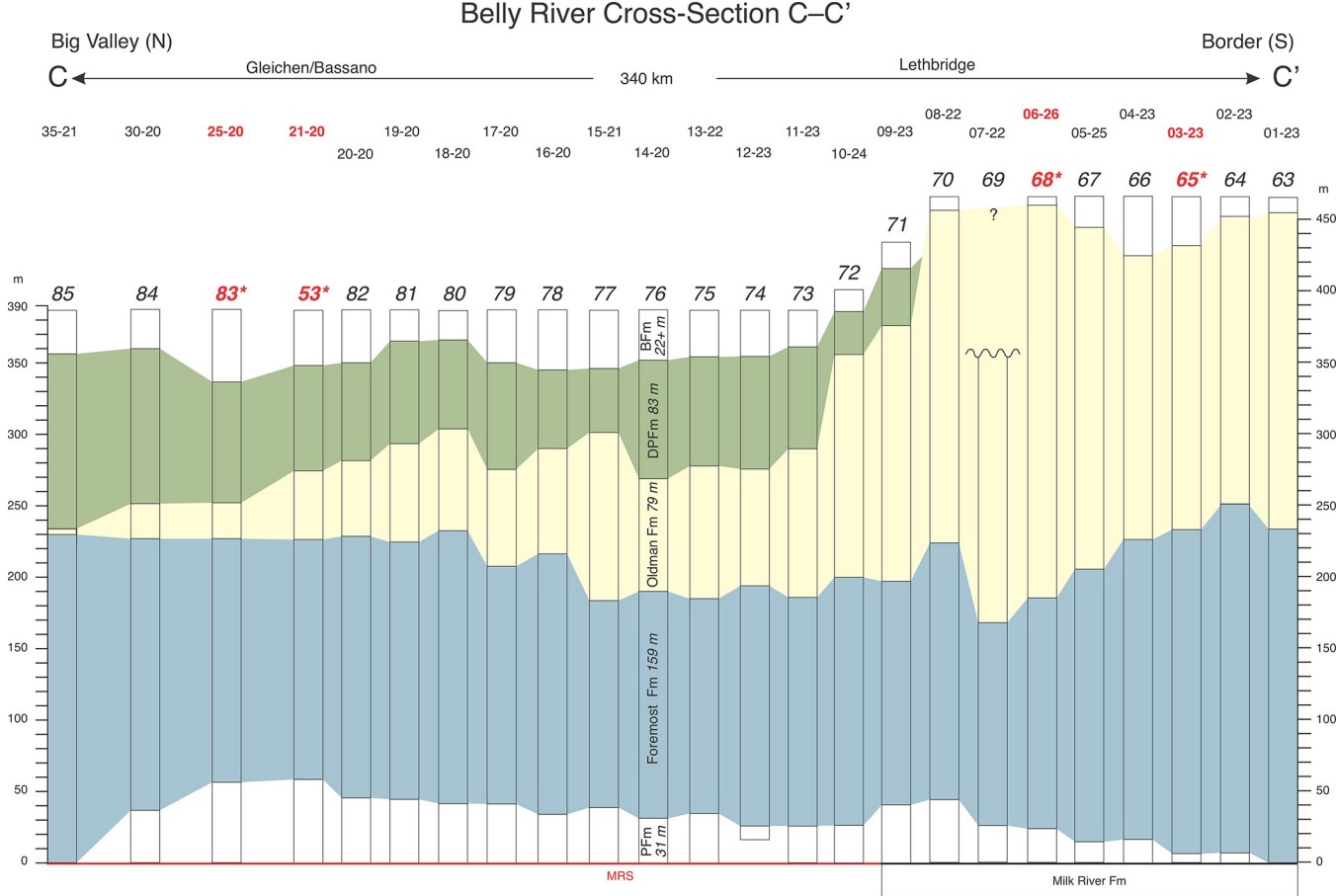

**Fig 11. Stratigraphic cross-section C–C'.** Cross-section C-C' is based on 24 well logs (Fig 1, Table 1, S3 Fig) and shows the extreme thickening of the Oldman Fm to the southwest, which underscores the presence of its unique, southwesterly source area. Note the southward termination (depositional edge) of the Dinosaur Park Fm near Lethbridge (Fig 1). Southward across the international border in this region, the Belly River Group correlates with the Two Medicine Formation.

in the subsurface and to identify and map the OFm-DPFm contact, which they interpreted as a regional discontinuity (ODPD). Specifically, they recognized a high gamma-ray log signature in the Oldman Fm followed by an abrupt leftward gamma-ray deflection (drop) upward across the base of the Dinosaur Park Fm as a unique feature that defines the OFm-DPFm contact and discontinuity in well logs from that area. However, they did not fully assess the utility of the high gamma-ray response as a diagnostic feature of the Oldman Fm elsewhere in southern Alberta. They also did not trace the OFm-DPFm contact and discontinuity in the subsurface to the southeast or southwest corners of the province, where the Oldman Fm had been observed to thicken considerably in outcrop [1, 17].

Hamblin [4, 5] and Hamblin and Lee [6] also recognized the Dinosaur Park Fm, and redefined the Oldman Fm in the subsurface as a two-fold unit consisting of a lower interval of sandstone(s) (their "Comrey Member sandstone") overlying the Taber Coal zone (and its equivalents), and an upper interval of mudstones with a relatively high gamma-ray response (their "upper siltstone member"). Hamblin [4] presented a gamma-density log pair from a well near Gleichen (see our Fig 3 as an example of a log pair from that area) as characteristic of the Oldman Fm's subsurface features across southern Alberta and north to Twp 45 and utilized

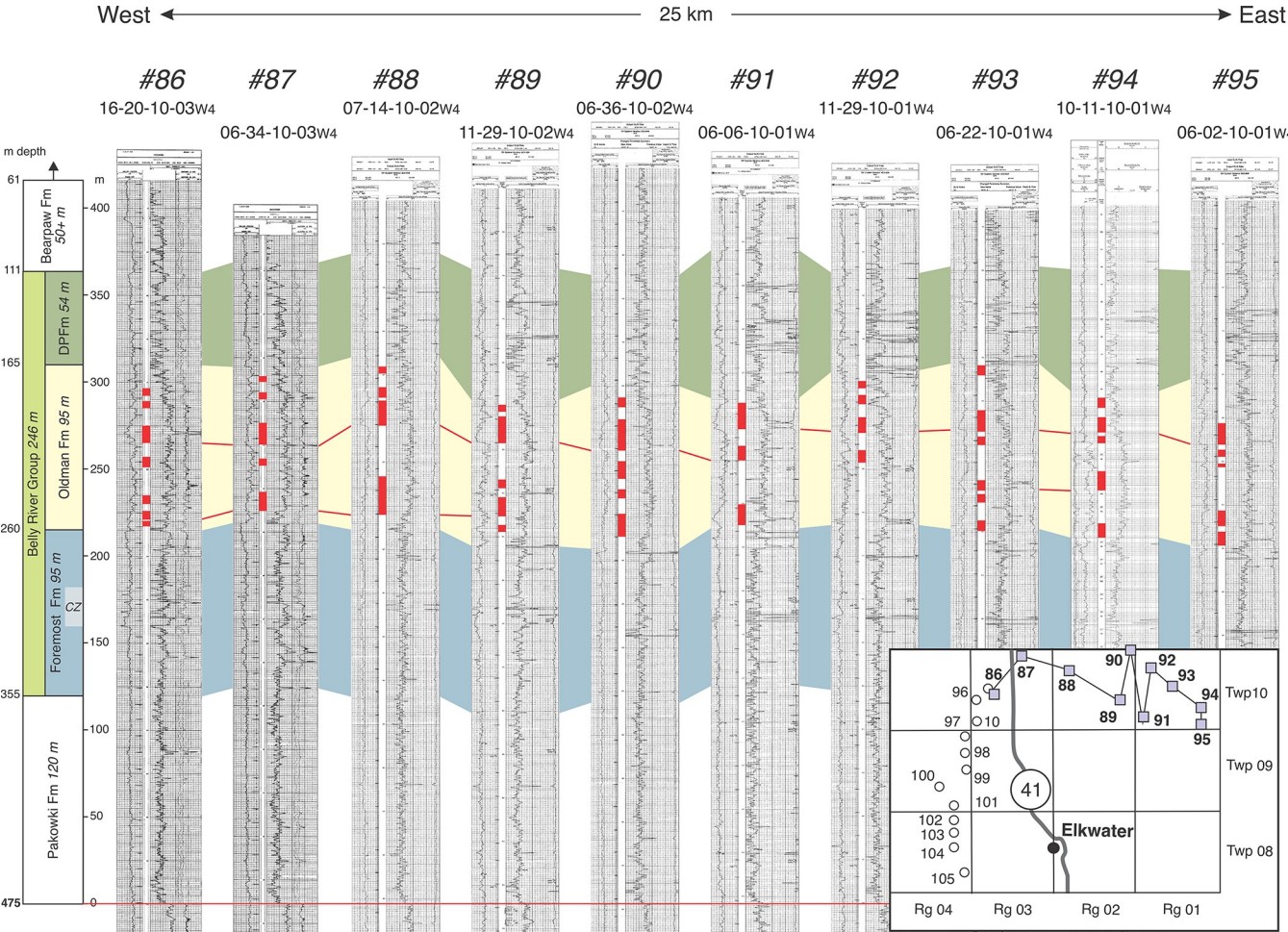

**Fig 12. High resolution east-west stratigraphic cross-section at Twp 10 in southeast Alberta.** Based on logs from 10 wells (Fig 1, Table 1, S4 Fig), the cross-section documents the presence of numerous alluvial sandstone bodies (red bars in well logs) throughout the Oldman Fm. Maximum e–w distance of ss-body correlation in this area is ~16 km (red lines), making regional correlation of the Comrey Sandstone or a "Comrey sandstone zone" unlikely.

gamma-ray-porosity log pairs to identify and trace the formation in the subsurface. Hamblin [4] showed the Oldman Fm having a thickness of about 50 m or less north of Twp 22 but provided no subsurface documentation south of Twp 19 (his Figs 3–6). Subsequent studies have relied on the data provided by Eberth and Hamblin [1] and Hamblin [4] to define the FFm-OFm and OFm-DPFm contacts [e.g., 7–9].

Whereas the formational picks and interpretations of thickness for the Oldman Fm as presented by Eberth and Hamblin [1] and Hamblin [4] are accepted here for areas of Alberta north of Twp 19, it is clear based on data gathered in this report that the subsurface characterization of the Oldman Fm in southernmost Alberta (south of Twp 19) as described by those authors remains limited and requires modification. Specifically, previous studies neither describe nor depict the subsurface (well log) features that characterize the greatly thickened Oldman Fm in southeastern and southwestern Alberta. The absence of an adequate subsurface characterization for the Oldman Fm in those areas has important implications for delineating the geometries and correlations of the formations that make up the BRG across southern Alberta and beyond the limits of the province.

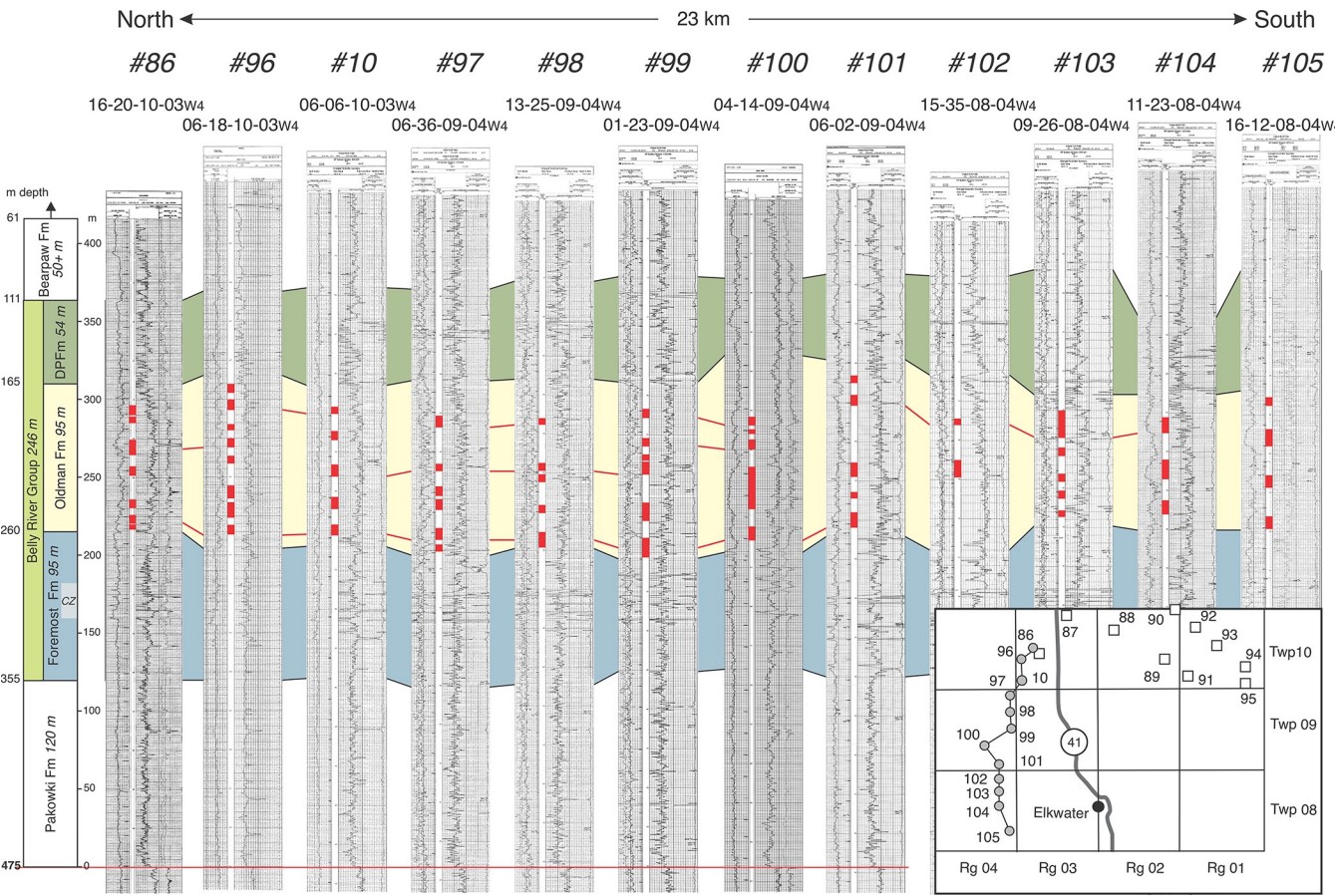

**Fig 13. High resolution north-south stratigraphic cross-section at Rg 04 in southeast Alberta.** Based on logs from 12 wells (Fig 1, Table 1, S4 Fig), this cross-section documents numerous alluvial sandstone bodies (red bars in well logs) throughout the Oldman Fm. Similar to the pattern shown in Fig 12, sandstone bodies show a maximum lateral extent of ~10 km, making regional correlation of the Comrey Sandstone or identification of a "Comrey sandstone zone" unlikely.

## Characterizing the Oldman Formation in the subsurface

Although the Oldman Fm is easily differentiated in outcrop from overlying and underlying formations by a large variety of sedimentological data [e.g., 1, 17], the unit has been difficult to distinguish from other formations using geophysical data alone [18]. An examination of well logs from southern Alberta in this study, specifically the southeastern and southwestern corners of the province, confirms that the most consistent feature that characterizes the Oldman Fm in the subsurface—and thus differentiates the three formations of the Belly River Group—is a relatively high gamma-ray response in Oldman Fm mudstones (an increase of ~10–30 API) compared to mudstones in the immediately underlying portions of the Foremost and overlying Dinosaur Park formations. Although relatively high gamma-ray responses may occur within single mudstone beds in the non-marine portions of the Foremost and Dinosaur Park formations, multi-meter successions of mudstones with similarly high gamma-ray responses remain absent. However, as the Foremost and Dinosaur Park formations transition down- and up-section, respectively, into marine shales of the Pakowki and Bearpaw formations, extensive intervals with high gamma-ray responses are re-established (Fig 3).

Well log data from six reference wells (Figs 3–5), many of which are tied to nearby exposures, show that as the Oldman Fm thickens toward the international border in both the

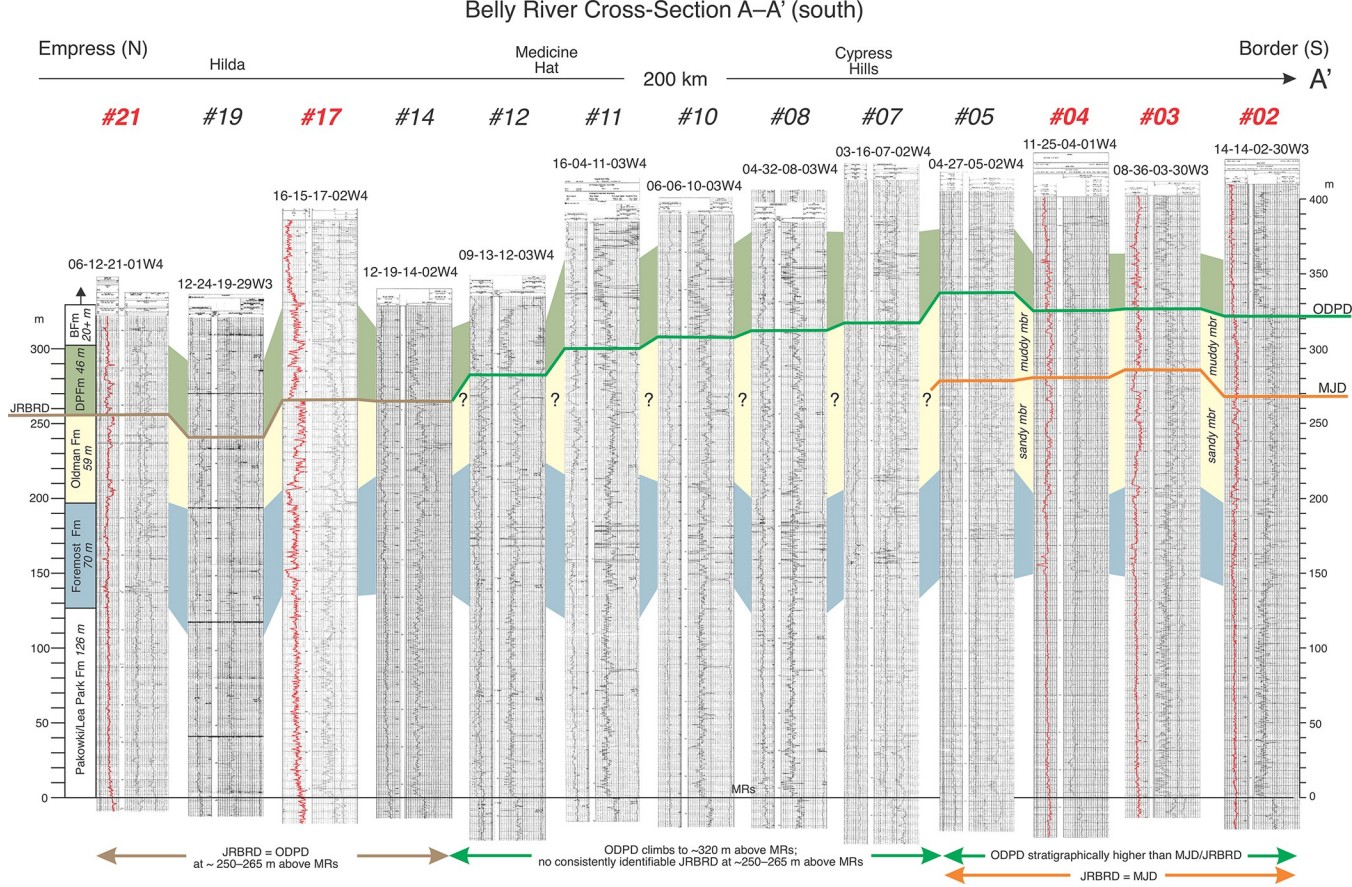

**Fig 14. Stratigraphic model depicting the relationship between the Oldman-Dinosaur Park discontinuity (ODPD) and the mid-Judith discontinuity (MJD).** The model is based on data and correlations presented here from cross-section A–A' (Fig 1, Table 1, S1 Fig) and differs from that of Rogers et al. [9] in that the ODPD and MJD are litho- and chronostratigraphically separate in southeastern-most Alberta due to up-section thickening of the Oldman Formation and thinning of the Dinosaur Park Fm. More specifically, in southeastern-most Alberta, the ODPD is positioned 320 or more meters above the MRs, whereas the MJD is positioned ~270 m above the MRs. North of Twp 12 in southeastern Alberta, however, the ODPD maintains a consistent position at about 240–260 above the MRs, thereby allowing the recognition of the combined Judith River-Belly River discontinuity (JRBRD) as originally proposed by Rogers et al. [9]. Well log numbers relate to positions shown in Fig 1.

southwest and southeast regions of the province, a high gamma-ray response is maintained throughout the formation. In each well log there are multiple intervals of mudstones with gamma-ray responses that are higher overall than those in the Foremost and Dinosaur Park formations. However, this pattern is made complex by the presence of multiple sandstone bodies throughout the Oldman Fm, many of which that show low gamma-ray responses like those in the Foremost and Dinosaur Park formations (Figs 4 and 5). These observations underscore that the relatively higher gamma-ray response that characterizes the Oldman Fm is mostly limited to the formation's mudstones, and that gamma-ray comparisons between the Foremost, Oldman and Dinosaur Park formations should, accordingly, focus on their mudstone units. Unfortunately, this point was only briefly touched on by Eberth and Hamblin [1, p. 183], and has subsequently been overlooked by subsequent researchers who have based their understanding of the Oldman-Dinosaur Park formational contact in the subsurface on well logs from north of Twp 19. Those logs show a clear dichotomy between a single interval of high-gamma mudstones in the Oldman Fm overlain by a prominent low gamma-ray response in sandstones that characterize the base of the Dinosaur Park Fm (Fig 3).

Whereas the Oldman Fm in subsurface can be characterized by the higher gamma-ray response of its mudstones (and some of its sandstones) relative to those in the Foremost and Dinosaur Park formations, its basal contact on the Foremost Fm is typically marked by an upward-fining (fluvial) sandstone(s) that marks the transition from carbonaceous mudstones, coals, and/or shallow-marine sandstones of the Foremost Fm to the high gamma-ray-response alluvial mudstones of the Oldman Fm [4] (Figs 3, 5–7). Similarly, following the convention first proposed by Eberth and Hamblin [1, p. 183], the Oldman-Dinosaur Park formational contact is placed at the base of the first sandstone that fines upward into one or more mudstone successions in the Dinosaur Park Fm with gamma-ray responses lower than those in the underlying Oldman Fm. As described by Hamblin [4], the contact with the overlying Dinosaur Park Fm is easily picked north of Twp 19 where the Oldman Fm is quite thin (~50 m thick, Fig 3). In the southeastern and southwestern corners of the province the Oldman Fm thickens considerably and placing the Oldman-Dinosaur Park formational contact requires consideration of the gamma-ray response within multiple mudstone successions between the Taber and Lethbridge coal zones.

Figs 6–8 show additional reference well logs from the southeast, central and northern regions of our field area that characterize variations in the gamma-ray response, thickness, and geometry of the three formations that make up the BRG. Measured section data from nearby outcrops that exhibit formational contacts are included and allow formational picks in well logs to be 'ground-truthed'. Specifically, Fig 6 includes two well logs in southcentral and southeast Alberta that can be directly correlated to outcrops on the Bow and South Saskatchewan rivers that exhibit the FFm-OFm and OFm-DPFm contacts. Fig 7 includes a well log from Dinosaur Provincial Park that can be correlated to the type outcrop section of the Dinosaur Park Fm near Iddesleigh (along the Red Deer River) and that exhibits the OFm-DPFm and DPFm-BFm contacts. Lastly, Fig 8 includes two well logs from the northeastern portion of our field area that can be correlated with fossiliferous exposures of the Oldman and Dinosaur Park formations at Muddy Lake, Saskatchewan (Twp 39, Rg 23W3), and that exhibit the FFm-OFm and OFm-DPFm contacts.

## Regional cross-sections

Regional cross-sections were assembled using 85 gamma-porosity log pairs (Table 1, S1–S3 Figs). These delineate Belly River Group formational geometries across southern Alberta (Figs 9–11). Although each cross-section is based on data from tens of well logs, individual logs are not shown in the cross-sections due to image-resolution constraints. Instead, the cross-sections exhibit color-coded formational picks and thicknesses at each well location, which are projected between locations.

Cross-section A–A' (Figs 1 and 9) runs ~400 km north-south along the Alberta-Saskatchewan border from Provost to the international border. Based on 35 well logs (Fig 1, Table 1, S1 Fig), it provides a strike section through the more distal portion of the Belly River Group clastic wedge at the eastern edge of our field area. From north to south the Oldman Fm mostly thickens stratigraphically up-section. At its most northern extent (Twp 40) the Oldman Fm is essentially a 10-m-thick marker bed that continues to thin and become unrecognizable farther north. At Twp 36, it is 25–35 m thick and maintains that thickness south to Twp 23 where it thickens sharply to more than 50 m. It maintains a thickness of 50–70 m farther south to Twp 11, where it thickens sharply to more than 80 m. From Twp 11 south to the international border the Oldman continues to thicken gradually, reaching a maximum of 126 m at Twp 02. From Twp 05 to the international border there is evidence for a two-fold subdivision of the OFm into a lower sandstone dominated interval and an upper mudstone dominated interval.

In contrast to the Oldman Fm, both the Foremost and Dinosaur Park formations in cross-section A–A' exhibit minor stratigraphic variations in thickness. The Foremost Fm exhibits a total stratigraphic thickness of ~70–80 m at Twp 40, thickens to ~100 m at Twp 22, and then thins gradually to 50–60 m south of Twp 04, near the international border. The Dinosaur Park Fm exhibits a more consistent southward-thinning trend with a stratigraphic thickness of 50–70 m from Twp 40 south to Twp 07, and then thinning to 30–40 m from there to the international border. The FFm-OFm contact remains essentially flat relative to the MRs throughout its extent in cross-section A–A', with a height above the MRs that ranges from 190–225 m. Its highest stratigraphic occurrence above the MRs is limited to the north (Twps 30–40) where it begins to pinch out, suggesting that the Foremost Fm thickens slightly up-section in that direction.

Cross-section B–B' (Figs 1 and 10) runs ~230 km west-east in Twp 21 from Calgary to the Alberta-Saskatchewan border near Empress. It is based on 28 well logs (Fig 1, Table 1, S2 Fig), reveals a dip section through the Belly River clastic wedge in southern Alberta, and exhibits pronounced eastward-stratigraphic-thinning of both the Foremost and Dinosaur Park formations as the BRG wedge extends farther into the basin. Similarly, the FFm-OFm contact drops gradually from east to west relative to the MRs reflecting the overall proximal-distal thinning of the entire wedge. Conversely, the Oldman Fm maintains a relatively constant thickness (40–60 m) and shows a slight stratigraphic thickening (up to 70 m) only in its most eastern locations (Rg 06–01).

Cross-section C–C' (Figs 1 and 11) extends for 340 km north-south from Big Valley to near the international border in southwestern Alberta. It is based on 24 well logs (Fig 1, Table 1, S3 Fig) and exhibits a strike section through a more proximal portion of the Belly River Group clastic wedge at the western edge of the field area. At Twp 35 and locations farther north, the Oldman Fm is a four-meter-thick marker bed that is difficult to identify in well logs; it eventually disappears north of Twp 45, marking the point where the Dinosaur Park and Foremost formations amalgamate and become indistinguishable. South of Twp 35 to the international border, the Oldman Fm thickens stratigraphically and mostly up-section. At Twp 10 it thickens sharply to ~160 m. At Twp 8 it thickens to more than 230 m, completely replaces the Dinosaur Park Fm, and is overlain by the Bearpaw Fm. From there south to the border it maintains a thickness of ~200 m whereas the Foremost Fm continues to expand down-section, eventually amalgamating with the underlying Milk River Fm at Twp 01. Like cross-section A–A' the FFm-OFm contact remains flat relative to the MRs with an average height of ~200 m above the MRs. However, the contact can exhibit a large amount of variation in stratigraphic position south of Twp 08. The Dinosaur Park Fm pinches out sharply at Lethbridge (Twp 08–09), an observation previously made by Hamblin [5].

## Discussion

### Updating the Belly River Group stratigraphic model

Data presented here support and add detail to the longstanding Belly River stratigraphic model [1, 4–7]. The existing model consists of three distinct stages in sediment supply and distribution, and shoreline orientation through the Belly River Group (Fig 2B). Each stage relates exclusively to one of the three formations that make up the BRG. The Foremost Fm reflects the earliest stage and is characterized by widespread western-to-northwestern sediment sources in the British Columbia cordillera supplying extensive north-south oriented shorelines that prograded cyclically to the east and ultimately formed stacked successions of upward-coarsening parasequences across southern Alberta [3, 6, 24]. The Oldman Fm records a second stage of stratigraphic evolution and is a mostly non-coaly nonmarine unit sourced primarily from

northwestern Montana and from limited regions of southwestern Alberta and southeastern British Columbia. Oldman Fm sediments prograded rapidly to the east and northeast across peat-forming wetlands of the uppermost Foremost Fm. Oldman Fm sediments ultimately prograded into central Saskatchewan where they recorded the maximum extent of the BRG wedge, as well as the maximum regression of the Pakowki/Lea Park sea [1, 4]. The third stage in the evolution of the BRG wedge records a variety of lithologic, mineralogic, sedimentary, geophysical, paleontological, and taphonomic differences between the Oldman and overlying Dinosaur Park formations, reflecting a significant reorganization in sediment supply and distribution across southern Alberta during the onset of transgression of the Bearpaw Sea (Fig 2B). Eberth and Hamblin [1] and others [4–7] interpreted the reorganization as due to the onset of tectonic rebound and unroofing in the proximal foredeep of the central cordillera (central British Columbia), an event that supplied a large volume of sediments and initially overwhelmed accommodation created during eustatic sea-level rise (Bearpaw transgression) and tectonic adjustments in southern Alberta. They concluded that Dinosaur Park Fm sediments expanded toward the southeast during transgression, gradually overstepping Oldman Fm wedge and expanding into northern-most Montana.

Data presented here support the existing three-stage tectono-stratigraphic model for the Belly River Group, but more clearly document the unique geometries and well-log signatures of the Oldman and Dinosaur Park formations that are important elements of the model. Specifically, the dramatic southwestward thickening of the Oldman Fm and overall higher gamma-ray response of the formation's fine-grained sediments confirms the presence of a geographically unique source area in northwestern Montana and southwestern Alberta. The data also document the development of a widespread and broadly isochronous horizon of carbonaceous shales and coals at the top of the Foremost Fm (Taber coal zone), upon which the Oldman Fm alluvial sediments were deposited. Regarding the Dinosaur Park Fm, the depositional edge of the formation can now be projected as a nnw–sse trending feature across southern Alberta (Fig 1). The southward thinning of the Dinosaur Park Fm is also more clearly documented relative to the thickening of the Oldman Fm, and thus further confirms that these two units were sourced from separate regions of the cordillera throughout their histories.

## Revising Oldman Formation lithostratigraphy and nomenclature

The data presented here allow us to clarify, update, and revise aspects of Oldman Fm stratigraphic nomenclature that have evolved during the past quarter-century. Eberth and Hamblin [1] modified the original definition of Oldman Fm [17] when they assigned the uppermost portion of the original OFm in the region surrounding Dinosaur Provincial Park to a new stratigraphic unit, the Dinosaur Park Fm, based on a variety of sedimentological, geophysical and paleontological criteria. Hamblin [4] then subdivided the Oldman Fm. He recognized two informal members making up the formation north of Twp 19: the lower "Comrey Member sandstone" and the "upper siltstone member". In his Figs 4, 6, 9 and 11, he consistently shows the base of his "Comrey Member sandstone" as more or less coincident with the base of the Oldman Fm on, or up to 2 m above the Taber coal zone, and 210–220 m above the Milk River shoulder.

However, outcrop data from southeastern Alberta along the Milk River canyon area (Fig 5) show that the base of the type Comrey Sandstone—originally named for the abandoned Comrey settlement near Manyberries (Twp 2, Rg 6; [17, 25])—lies ~60 m or more above the Taber coal zone in the middle of the Oldman Fm, and ~260–265 m above the Milk River shoulder datum. Using these data, Eberth [7] applied the term "Comrey sandstone zone" to mineralogically mature sandstones in the middle of the Oldman Fm throughout southeastern Alberta,

and proposed them to be stratigraphically equivalent to the type Comrey Sandstone exposed along the Milk River canyon, just south of Comrey.

Eberth [7] recognized the same sandstone zone recognized by Hamblin (1997a) on or just above the Taber coal zone throughout southern Alberta in logs and outcrop but referred it to the "Herronton sandstone zone" after the hamlet of Herronton, which lends its name to the productive, shallow-gas-field that produces mostly from this sandstone horizon southeast of Calgary. Eberth [7] assigned the Herronton sandstone zone to the top of the Foremost Fm due to its distinctive mineralogy (immature and bentonitic fine-to-medium-grained sandstones) and sedimentology (locally incised multistoried high-sinuosity paleochannels characterized by clean sandstone and inclined heterolithic stratification [IHS]). Data presented here from numerous locations along the Bow and South Saskatchewan Rivers and the Milk River canyon area of southeastern Alberta (Figs 5 and 6) indicate the presence of a well-developed multimeter thick sandstone zone, often consisting of IHS paleochannel deposits, on or just above the Taber coal zone, and thus equivalent to the Herronton sandstone zone of Eberth [7].

A comparison of the interpretations of both Hamblin [4, Fig 14] and Eberth [7, Fig 3.9], as well as a review of their sedimentological data, show that the "Comrey Member sandstone" of Hamblin [4] and "Herronton sandstone zone" of Eberth [7] refer to the same sandy gas-producing stratigraphic interval that rests on, or slightly above, the Taber coal zone. For reasons of nomenclatural clarity, and to maintain consistent reference to and accurate stratigraphic placement of the type Comrey Sandstone in southeastern Alberta as originally presented by Troke [25], we propose that the sandy, IHS-rich interval that is positioned at the base of the Oldman Fm on or just above the Taber coal zone be henceforth referred to as the "Herronton sandstone zone" as proposed by Eberth [7] and in line with industry nomenclatural practice. Although future work may reveal that the Herronton sandstones are mineralogically unlike those higher in the Oldman Fm (as suggested by Eberth [7] and supported by the data of Eberth and Hamblin [1, their Fig 6], it is proposed here that the Herronton sandstone zone be included, for the time being, at the base of the Oldman Fm. This follows lithostratigraphic and historical precedents set by most researchers who have sought to distinguish the Foremost and Oldman (Pale Beds) formations based on paleoenvironmental criteria (Foremost paralic facies versus Oldman alluvial facies). More significantly, assigning the Herronton sandstone zone to the base of the Oldman Fm continues to allow for meaningful comparisons with the large amount of previously published and archived Oldman Fm subsurface data (e.g., picks, geometries, and isopach-calculations) that are based on the inclusion of the Herronton sandstone zone in that formation [3–6, 8].

The present study also demonstrates that the Oldman Fm thickens substantially to the southeast and southwest, stratigraphically extending the Oldman Fm (and its multiple mudstone and sandstone successions) up-section, well above the "upper siltstone member" as defined and depicted by Hamblin [4]. By Twp 13 (Fig 6) and farther south (Fig 5), the stratigraphically thickened Oldman Fm exhibits multiple stratigraphic intervals dominated by mudstone and sandstones, and no single interval can be deemed equivalent to the "upper siltstone member". Accordingly, because that term ceases to have any lithostratigraphic meaning south of Twp 16 and is deemed here to be misleading—the "upper siltstone" is equivalent to the lowest portions of the Oldman Fm in southern-most Alberta—it is recommended that the term be dropped altogether in favor of the simpler and more inclusive "Oldman Formation".

### Is there a "Comrey sandstone zone" in southern Alberta?

The Comrey Sandstone was defined and described by Troke [25] in a study of its type exposures in the walls of the Milk River canyon in southeastern Alberta (Twp 02, Rg 06). It occurs

stratigraphically in the middle of the Oldman Fm, ~60 m above the base of the formation (Fig 5) and has been successfully utilized by vertebrate paleontologists as a local marker with which to divide the Oldman Fm into lower and upper intervals in the Manyberries region of south-eastern Alberta [26, 27]. Similarly, Eberth [7] proposed that the Comrey Sandstone may be part of a laterally continuous and stratigraphically identifiable sandstone zone across south-eastern Alberta based on informal observations of outcrop along the South Saskatchewan River, north of Medicine Hat. However, the existence of such a continuous sandstone zone was not tested with subsurface data, and thus remains unproven.

Well logs from cross-section A–A' (S1 Fig) document the presence of numerous well-developed fluvial sandstones throughout the Oldman Fm, including those with stratigraphic positions that are approximately equivalent to the type Comrey Sandstone (~260–265 m above the MRs). However, because of the ~10 km distance between wells in that cross-section, it is difficult to determine if sandstone bodies are laterally continuous between adjacent wells. Accordingly, a higher resolution study was undertaken in a smaller area along the A–A' transect (Twp 08–10, Rg 1–4) using data from 22 wells (Fig 1, Table 1, S4 Fig) with one-well-per-section spacing (Figs 12 and 13). This area was chosen because of the large number of available wells that include complete sections of the Oldman Formation, and because the location of the study area is midway between the Milk River canyon area and Oldman Fm exposures along the South Saskatchewan River north of Medicine Hat—the areas that were the basis for the original hypothesis of Eberth [7]. These high-resolution cross-sections (Figs 12 and 13) reveal that although sandstone bodies are abundant throughout the Oldman Fm, there is neither a distinct (e.g., uniquely thick) nor a single, laterally continuous sandstone horizon at the expected stratigraphic position of the type Comrey Sandstone (or anywhere else in the Oldman Fm). The maximum lateral continuity of sandstone bodies in east-west or north-south directions is 16 km (Figs 12 and 13). Based on the absence of unique or laterally continuous sandstones, and the absence of evidence for lateral continuity of the type Comrey Sandstone itself beyond its outcrops south of Manyberries, the hypothesis of a discrete and traceable "Comrey sandstone zone" throughout southern Alberta that is laterally continuous with the type Comrey Sandstone is rejected here.

## Clarifying and refining the Judith River-Belly River "problem"

The data presented here can be used to clarify and refine results of a recent study of the Judith River-Belly River clastic wedge by Rogers et al. [9]. That study seeks to resolve lithostratigraphic correlations and nomenclatural differences between the Judith River Formation (USA) and Belly River Group (Canada) and proposes the physical continuity and isochroneity of two previously recognized discontinuities (the mid-Judith discontinuity in Montana [MJD] and the Oldman-Dinosaur Park discontinuity in Alberta [ODPD]). It establishes the single term, Judith River-Belly River discontinuity (JRBRD), to refer to the widespread north-south expression of the combined surfaces. The study further emphasizes the consistent stratigraphic thickness of ~250–265 m between the MRs datum and the JRBRD throughout the field area and, using the U-Pb geochronology of Ramezani et al. [11], assigns an age of ~76.3 Ma to the JRBRD. Lastly, the study builds on the work of Rogers et al. [14] and interprets the combined discontinuities as reflecting a widespread pulse of accommodation resulting from tectonically induced subsidence and the onset of transgression of the Bearpaw Sea at ~76.3 Ma.

Data and interpretations presented here differ somewhat from those of Rogers et al. [9] in presenting a more nuanced picture of the two discontinuities and their geographic relationships. Whereas the Oldman-Dinosaur Park formational contact and discontinuity (ODPD) conforms to the pattern described by Rogers et al. [9] throughout the central and northern

portions of our field area (north of Twp 12) the ODPD is neither continuous nor isochronous with the mid-Judith discontinuity (MJD) in the southeastern and southwestern corners of Alberta south of Twp 12 (Figs 9, 11, 14). Specifically, data presented here demonstrate that at Twp 12 and locations farther south to the international border in southeastern Alberta, the ODPD climbs significantly higher stratigraphically and, ultimately, becomes established ~320 m above the Milk River shoulder, ~50 m higher than the MJD (Figs 9 and 14). Similarly, in southwestern-most Alberta at Lethbridge (Twp 10) the ODPD rises sharply higher stratigraphically and becomes established ~357 m above the Milk River shoulder (Fig 11). South of Lethbridge at Twp 08 the Dinosaur Park Fm pinches out altogether and the ODPD ceases to exist (Fig 11; cf. Hamblin [5]).

Differences in interpretation between Rogers et al. [9] and the present study are understandable. First, Rogers et al. [9] did not include locations farther west than Rg 12 in their study. Accordingly, they did not integrate the obvious and extreme southwestward thinning and pinch-out of the Dinosaur Park Fm near Lethbridge in their interpretations. Secondly, Rogers et al. [9] failed to integrate into their study the southeastern thickening of the Oldman Fm to the international border as originally described by Eberth and Hamblin [1] and later underscored by isopach (sediment dispersal) data [4–6, 8]. Accordingly, south of Twp 12 in southeastern Alberta and southwestern Saskatchewan, Rogers et al. [9] place many of their well log picks for the ODPD too low in section (their Figs 7, 8). Their misplaced picks likely also resulted from confusion as to how to identify the Oldman-Dinosaur Park formational contact (and thus the ODPD) in gamma-ray logs. As discussed above, Eberth and Hamblin [1] limited their application of the characteristic ODPD gamma-ray log signature to areas in and around Dinosaur Provincial Park (Twp 20, Rg 12) where the ODPD is commonly identified as occurring at the base of a sandstone with low gamma-ray response and sharply in contact with a limited stratigraphic interval of mudstones with high gamma-ray peaks (e.g., Figs 3 and 7). Because Eberth and Hamblin [1] did not emphasize the importance of assessing the gamma-ray responses in the multiple mudstone successions that occur between the Taber and Lethbridge coal zones as the Oldman Formation thickens to the south in Alberta, correct placement of the ODPD has remained problematic. Rogers et al. [9] followed the practice developed by Eberth and Hamblin [1] for identifying the ODPD in subsurface in and around Dinosaur Provincial Park and placed their ODPD picks throughout southern Alberta at the base of the first prominent sandstone that they recognized above the first prominent occurrence of a high gamma-ray response. In so doing, they failed to incorporate the complexity required to accurately identify the ODPD in a significantly thickened Oldman Formation in southeastern Alberta. Lastly, Rogers et al. [9] did not employ measured section outcrop data with which they could calibrate and test their interpretations. Accordingly, whereas they correctly correlated U-Pb ages of the MJD and ODPD in central Montana and at Dinosaur Provincial Park, respectively, and correctly linked two different stratigraphic expressions for the widespread tectonic event(s) that initiated distal foredeep subsidence and influenced the eustatic rise of sea level and marine transgression throughout the northern portion of the WIB, they failed to fully address the changes in BRG stratigraphic architecture just north of the international border.

It is proposed here that the results of Rogers et al. [9] need only be modified slightly by recognizing that the ODPD climbs in section south of Twp 12 (in southeastern Alberta and southwestern Saskatchewan) and that, from Twp 12 to the international border, the two discontinuities (MJD, ODPD) remain stratigraphically separate (Fig 14). In this context, Rogers et al.'s definition of the Judith River Belly River discontinuity (JRBRD) should be revised as representing a widespread isochronous (~76.3 Ma) surface that includes the entire MJD (north-central Montana, southeast Alberta, and southwest Saskatchewan), and that portion of

the ODPD that lies north of Twp 12 in southern Alberta and southwestern Saskatchewan. This revised definition necessarily calls attention to a ~60–80 km north-south gap or geographic zone between Twp 14 and Twp 05 in southeastern Alberta (Fig 14) where there exists little evidence for either the MJD or ODPD 250–265 m above the MRs, and thus, no direct evidence for the JRBRD. Here, the geographic gap is interpreted as reflecting a complex interaction between two distinct and regionalized forms of tectonic activity that occurred during the evolution of the combined Judith River-Belly River wedge: (1) the development of a subsidence-induced expansion surface in the Judith River Fm (mid-Judith discontinuity, MJD) of north-central Montana and (2) the tectonically induced rebound of the proximal foredeep in central British Columbia that resulted in the advance of the Dinosaur Park Fm clastic wedge across southern Alberta.

## Implications for vertebrate fossil occurrences

The complex internal stratigraphic architecture of the Belly River Group described here has implications for stratigraphic assignment of vertebrate fossils, especially in the Oldman and Dinosaur Park formations. For example, the reciprocal wedged-shaped geometries of the Oldman and Dinosaur Park formations across southern Alberta suggest that some fossil locations that occur in the lower one-half of the Dinosaur Park Fm at northern locations (e.g., Hilda, Sandy Point, Jenner, and Dinosaur Provincial Park) likely are time equivalent with fossil locations that occur in the upper one half of the Oldman Fm at south-eastern locations (e.g., Manyberries area). This possibility has been mentioned or discussed previously [26–28] based on taxonomic comparisons, and the hypothesis can now be tested more rigorously using high-resolution U-Pb CA-ID-TIMS dating of bentonites from each location [9, 11, 12].

Additionally, based on the relatively consistent stratigraphic distance between the Milk River shoulder and the FFm-OFm contact documented here (especially in eastern Alberta) fossil occurrences known to occur stratigraphically close to the FFm-OFm contact [29, 30] may prove to be close in age. This suggests broadly isochronous ages for other more widely dispersed vertebrate fossil occurrences from Milk River Canyon, Redcliff, Suffield, Hays/Rolling Hills, Jenner, Sandy Point, and Rapid Narrows, many of which remain unpublished.

## Summary and conclusions

This comprehensive subsurface study of the Belly River Group extends across all southern Alberta and thereby allows previously proposed interpretations of stratigraphic architecture based on smaller geographic areas and more limited stratigraphic intervals to be tested, refined, modified, or rejected. Principal among these is the geometry of the Oldman Fm relative to both the underlying Foremost Fm and the overlying Dinosaur Park Fm. Subsurface data presented here confirms thickening of the Oldman Fm to the south and principally to the southwest into Montana, and documents stratigraphic thickening of the Oldman Fm primarily up-section at the expense of the overlying Dinosaur Park Fm. It also underscores the increasingly younger, diachronous nature of the Oldman-Dinosaur Park formational contact to the south, and the widespread possibly isochronous nature of the Foremost-Oldman formational contact across southern Alberta. Lastly, as the Dinosaur Park Fm thins to the south and pinches out to the southwest on the Oldman Fm, the depositional edge of the formation is resolvable as a wnw–ese trending line, and low gamma/high porosity signatures characteristic of the Lethbridge coal zone continue south into the uppermost Oldman Fm.

Up-section stratigraphic thickening of the Oldman Fm requires that our understanding of the formation in the subsurface be modified and broadened in several ways. First, although the higher gamma-ray response of mudstone successions through the formation remains a reliable

means for identifying the formation in well logs and distinguishing it from the Foremost and Dinosaur Park formations it is also clear that there exist many sandstones with low gamma-ray responses within the Oldman Fm that are equivalent in response to those in the underlying and overlying formations. Thus, picking the top or the bottom of the formation requires careful consideration of the entire gamma-ray response pattern between the Taber and Lethbridge coal zones of the Belly River Group, an approach that has yet been applied consistently by students of BRG stratigraphy. Up-section thickening of the Oldman Fm also forces a reconsideration of two terms ("Comrey Member sandstone" and "upper siltstone member") that were previously applied in reference to a two-fold subdivision of the Oldman Fm north of Twp 19. Recognition that the type Comrey Sandstone occurs in the middle of the Oldman Formation in southeastern Alberta suggests that previous use of that term for the sandstone zone that forms the base of the Oldman Fm throughout large portions of southern Alberta is incorrect and should be replaced by the term Herronton sandstone zone. The Herronton sandstone zone is also recognized at the base of the Oldman Fm above the Taber coal zone in local outcrops along the Red Deer, Bow, South Saskatchewan, and Milk rivers of southern Alberta. The "upper siltstone member" was originally proposed for a stratigraphically limited interval (typically less than 50 m thick). However, in both southeastern and southwestern Alberta, where the formation thickens to more than 100 m or more, the term is neither stratigraphically accurate nor useful, and thus should be avoided.

Rogers et al.'s [9] correlation of discontinuities from both the Judith River Formation and the Belly River Group across the international border also requires reconsideration given the improved understanding of Belly River Group stratigraphic architecture provided here. It is clear now that different and regionalized tectonic influences on stratigraphic architecture were at work in the Judith River Formation and the Belly River Group during the onset of the Bearpaw transgression. In particular, the development of a subsidence-induced expansion surface in the Judith River Fm (mid-Judith discontinuity, MJD) versus a tectonically induced rebound of the proximal foredeep that resulted in a reorganization of sediment distribution systems across southern Alberta (the Oldman-Dinosaur Park discontinuity, ODPD). New U-Pb geochronologic data indicate that the initiation of these regionalized events was closely related in time, likely tectonically related, and thus correlable [11, 12, 14]. However, data presented here also show that the ODPD is locally diachronous and becomes significantly younger south of Twp 12 as the Dinosaur Park Fm overrides the Oldman Fm in a step-wise fashion. This pattern indicates that Judith River Belly River discontinuity—the term referring to the combined MJD and ODPD—consists of the MJD but only the oldest portions of the ODPD (north of Twp 12).

These data and modifications to our understanding of Belly River stratigraphic architecture will likely improve correlation of fossiliferous exposures in southern Alberta as well as correlations of the Belly River Group with the Judith River and Two Medicine formations in northern Montana.

## Supporting information

**S1 Fig. Annotated well logs (n = 35) used in cross-section A–A'.**
(PDF)

**S2 Fig. Annotated well logs (n = 28) used in cross-section B–B'.**
(PDF)

**S3 Fig. Annotated well logs (n = 24) used in cross-section C–C'.**
(PDF)

**S4 Fig. Well logs (n = 22) used in high-resolution cross-sections in Figs 12 and 13.**
(PDF)

## Acknowledgments

Divestco Geoscience Ltd. (Calgary) and staff are thanked for providing access to their raster and digital well log library, and for assistance in use of their digital platform (Energisite). Specifically, I thank Tania Uszacki (Product Manager) and Veronica Holmes for support that made this project possible. Ray Rogers was instrumental in revitalizing my interest in this project and incentivising the completion of research initiated in 1986. I thank him for the many interesting and stimulating discussions concerning discontinuities and upper Belly River stratigraphy discussed here. I thank Federico Fanti and David Evans for graciously sharing their Belly River outcrop data and opinions from the Milk River canyon area of southeastern Alberta. Field assistance was provided by Tyrrell Museum technical staff over many decades and additional support was supplied by the Tyrrell Museum Cooperating Society. Special thanks to Marty who shared in the day-to-day advances and setbacks inherent in developing a data-heavy manuscript.

## Author Contributions

**Conceptualization:** David A. Eberth.

**Data curation:** David A. Eberth.

**Formal analysis:** David A. Eberth.

**Investigation:** David A. Eberth.

**Methodology:** David A. Eberth.

**Project administration:** David A. Eberth.

**Resources:** David A. Eberth.

**Supervision:** David A. Eberth.

**Validation:** David A. Eberth.

**Visualization:** David A. Eberth.

**Writing – original draft:** David A. Eberth.

**Writing – review & editing:** David A. Eberth.

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
