## [Decision Letter · Decision Letter 0]

1 Aug 2023

PONE-D-23-14310Stratigraphic architecture of the Belly River Group (Campanian) in the plains of southern Alberta: Revisions and updates to an existing model, and implications for correlating dinosaur-rich strataPLOS ONE

Dear Dr. Eberth,

Thank you for submitting your manuscript to PLOS ONE. After careful consideration, we feel that it has merit but does not fully meet PLOS ONE’s publication criteria as it currently stands. Therefore, we invite you to submit a revised version of the manuscript that addresses the points raised during the review process.

We look forward to receiving your revised manuscript.

Kind regards,

Jürgen Kriwet

Academic Editor

PLOS ONE

Journal Requirements:

2. In your manuscript, please provide additional information regarding the specimens used in your study. Ensure that you have reported human remain specimen numbers and complete repository information, including museum name and geographic location. 

For more information on PLOS ONE's requirements for paleontology and archeology research, see https://journals.plos.org/plosone/s/submission-guidelines#loc-paleontology-and-archaeology-research

4. We note that you have referenced (Eberth, D.A., Evans, D.C., Ramezani, J., Kamo, S., Brown, C.M., Currie, P.J., Braman, D.R. [submitted]. Calibrating geologic strata, dinosaurs, and other fossils at Dinosaur Provincial Park (Alberta, Canada) using a new CA-ID-TIMS U–Pb geochronology.) which has currently not yet been accepted for publication. Please remove this from your References and amend this to state in the body of your manuscript: (ie “Bewick et al. [Unpublished]”) as detailed online in our guide for authors

5. We note that Figures 1 and 2 in your submission contain map/satellite images which may be copyrighted. All PLOS content is published under the Creative Commons Attribution License (CC BY 4.0), which means that the manuscript, images, and Supporting Information files will be freely available online, and any third party is permitted to access, download, copy, distribute, and use these materials in any way, even commercially, with proper attribution. For these reasons, we cannot publish previously copyrighted maps or satellite images created using proprietary data, such as Google software (Google Maps, Street View, and Earth). For more information, see our copyright guidelines: http://journals.plos.org/plosone/s/licenses-and-copyright.

a. You may seek permission from the original copyright holder of Figures 1 and 2 to publish the content specifically under the CC BY 4.0 license.  

Additional Editor Comments:

n/a

Reviewers' comments:

Reviewer's Responses to Questions

**Comments to the Author**

1. Is the manuscript technically sound, and do the data support the conclusions?

Reviewer #1: Yes

2. Has the statistical analysis been performed appropriately and rigorously? 

Reviewer #1: Yes

3. Have the authors made all data underlying the findings in their manuscript fully available?

Reviewer #1: Yes

4. Is the manuscript presented in an intelligible fashion and written in standard English?

Reviewer #1: Yes

5. Review Comments to the Author

Reviewer #1: In my opinion, the work is well written and of scientific quality in all the aspects, with data that support the reached conclusions. The paper deals with regional lithoestratigraphy, dealing with late Cretaceous units that are out of my knowledge, so I am not able to judge the relevance of the contributions.

My only concern deals with the boundaries that actually represent (at least in part) lateral facies changes. In Figure 2A, I understand should by some interfingering in the lower and upper boundaries of the Oldman Fm (similar to the lower boundary of the Foremost Fm, or the upper boundary of the DP Formation). Similarly, the concluding sentence in lines 590-591, "documents stratigraphic thickening of the Oldman Fm ... at the expense of the overlying DP Fm" should be clarified: Do you mean a lateral facies change? This aspect should be also clarified in Figures 9-10-11... both, in Figure caption and in presentation of data and discussion.

6. PLOS authors have the option to publish the peer review history of their article (what does this mean?). If published, this will include your full peer review and any attached files.

Reviewer #1: No

---

## [Author Response · Author response to Decision Letter 0]

25 Aug 2023

Response to Reviewers

Academic Editor review

1. I have revised the ms formatting and file names to bring them in line with PLOS style. I did use PACE to convert from my PDFs and check the figures. However, I found that the quality of the TIFFs created by that program was much poorer than the PDFs and original CorelDraw images. I custom transformed the images to TIFFs myself and was much happier with the results (although they are still somewhat degraded). I strongly recommend that PLOS consider allowing PDFs as text figures (along w TIFF and EPS). PDFs are industry standard and EPS files are somewhat unreliable and are being phased out in the graphics community. TIFFs tend to be pixelated due to the transformation algorithm. Thanks for considering the request.

2. There are no specimens — rock, fossil, or otherwise — used in this study. Thus, further information is not required. No field permits were required at the time that field work was conducted (1980s). Accordingly, I have added a section at the end of the ms that states: “No permits were required for the described study, which complied with all relevant regulations.”

3. I have removed mention of archived field notes in the Methods section. The measured sections that are depicted here (Figs 4–8) are standard sedimentary geology logs that were drafted originally in the field, and thus require no access to field notes for further explanation. Similarly, there are no ethical or legal issues surrounding the data, nor any need to upload written field notes as an additional Supporting Information file.

4. The ms by Eberth, D.A., Evans, D.C., Ramezani, J., Kamo, S., Brown, C.M., Currie, P.J., Braman, D.R, is now published (Canadian Journal of Earth Sciences, 2023). Reference to that publication has been updated throughout the ms.

5. Figure 1 is an original map drafted by me for this ms and contains elements (e.g., Twp-Rg designations, rivers, city locations, my own proposed Belly River outcrop limits, cross-section and well locations) that are either unique to this image or clearly reside in the public domain. In short, Figure 1 is not derived from any one pre-existing image. I replaced figure 2B with a new image that contains new scientific information and was drafted in the same style as other images in this ms. 

6. Captions are included for Supporting Information files.

7. The reference list includes no retractions or new references. It does, however, include updated information for the publications by Eberth et al. (2023) and Rogers et al. (2023), which are now published.

Peer Reviewer review

The peer reviewer raised only a single issue (#5). The reviewer is curious about the lateral relationships of the Oldman and Dinosaur Park formations to one another. More specifically: should the contact, where it is diachronous, be described and expressed in images as interfingering? 

This is an excellent observation and comment, and one that many of us have puzzled over (on and off) during the past 30 years. Well log data clearly document interfingering contacts between marine shales and the Belly River Group (as shown in Figure 2A), but such interfingering has never been clearly observed or documented between the group’s non-marine units where facies differences are much more subtle. Documentation of interfingering would likely require a comparative geochemical analysis of mudstones from many outcrops conducted over many kilometers in order to test for differences in the fine-grained facies of the Oldman and Dinosaur Park formations— an undertaking that would prove expensive and time consuming, and would not be guaranteed to provide important new information.

Whereas it seems entirely reasonable to assume that there should be some degree of mixing and interfingering at the boundaries of these two clastic wedges, we remain hesitant to make that claim or represent it in our figures for fear of misrepresenting the degree to which both wedges are competing at their edges. For example, the subsurface data presented here (e.g., Fig 9, 11) clearly suggest a step-wise relationship between the two wedges, especially at locations along the Alberta-Saskatchewan border. Such step-wise relationships suggest the presence of local areas where the lateral extent of interfingering may be quite limited. Similarly, there are extensive areas between the “steps” where the contact remains more-or-less flat. Together these data suggest that a depiction of interfingering may incorrectly represent the true nature of the interaction of these wedges. I have modified Fig 2A to better highlight the step-like nature of this contact in southeastern Alberta.

In summary, we have opted for use of “shazam” lines in our graphics where we have evidence of interfingering (Foremost-Lea Park and Dinosaur Park-Bearpaw contacts), and avoid their use where such evidence is currently lacking.

To my knowledge there are no other outstanding criticisms of this ms.

---

## [Editor Report · Decision Letter 1]

18 Sep 2023

Stratigraphic architecture of the Belly River Group (Campanian, Cretaceous) in the plains of southern Alberta: Revisions and updates to an existing model, and implications for correlating dinosaur-rich strata

PONE-D-23-14310R1

Dear Dr. Eberth,

We’re pleased to inform you that your manuscript has been judged scientifically suitable for publication and will be formally accepted for publication once it meets all outstanding technical requirements.

Kind regards,

Jürgen Kriwet

Academic Editor

PLOS ONE
---

## [Editor Report · Acceptance letter]

25 Sep 2023

PONE-D-23-14310R1 

Stratigraphic architecture of the Belly River Group (Campanian, Cretaceous) in the plains of southern Alberta: Revisions and updates to an existing model and implications for correlating dinosaur-rich strata 

Dear Dr. Eberth:

I'm pleased to inform you that your manuscript has been deemed suitable for publication in PLOS ONE. Congratulations! Your manuscript is now with our production department. 

Kind regards, 

on behalf of

Dr. Jürgen Kriwet 

Academic Editor

PLOS ONE